# The FireWork v2.0 air quality forecast system with biomass burning emissions from the Canadian Forest Fire Emissions Prediction System v2.03

Jack Chen[1], Kerry Anderson[2+], Radenko Pavlovic[3], Michael D. Moran[1], Peter Englefield[2], Dan K. Thompson[2], Rodrigo Munoz-Alpizar[3], and Hugo Landry[3]

[1] Air Quality Research Division, Environment and Climate Change Canada, Ontario, Canada
[2] Canadian Forest Service, Natural Resources Canada, Alberta, Canada
[3] Air Quality Modelling Applications Section, Environment and Climate Change Canada, Quebec, Canada
[+]Emeritus

*Correspondence to*: Jack Chen (jack.chen@canada.ca)

**Abstract.** Biomass burning activities can produce large quantities of smoke and result in adverse air quality conditions in regional environments. In Canada, Environment and Climate Change Canada's (ECCC) operational FireWork (v1.0) air quality forecast system incorporates near-real-time biomass burning emissions to forecast smoke plumes from fire events. The system is based on the ECCC operational Regional Air Quality Deterministic Prediction System (RAQDPS) augmented with near-real-time wildfire emissions using inputs from the Canadian Forest Service's (CFS) Canadian Wildland Fire Information System (CWFIS). Recent improvements to the representation of fire behaviour and fire emissions have been incorporated into the CFS Canadian Forest Fire Emissions Prediction System (CFFEPS) v2.03. This is a bottom-up system linked to CWFIS in which hourly changes in biomass fuel consumption are parameterized with hourly forecasted meteorology at fire locations. CFFEPS has now also been connected to FireWork. In addition, a plume-rise parameterization based on fire energy thermodynamics is used to define the smoke injection height and the distribution of emissions within a model vertical column. The new system, FireWork v2.0 (FireWork-CFFEPS), has been evaluated over North America for July-September 2017 and June-August 2018, both periods when western Canada experienced historical levels of fire activity with poor air quality conditions in several cities as well as other fires affecting northern Canada and Ontario. Forecast results were evaluated against hourly surface measurements for the three pollutant species used to calculate the Canadian Air Quality Health Index (AQHI), namely $PM_{2.5}$, $O_3$, and $NO_2$, and benchmarked against the operational FireWork v1.0 system (FireWork-Ops). This comparison shows improved forecast performance and predictive skills for the FireWork-CFFEPS system. Modelled fire plume injection heights from CFFEPS based on fire energy thermodynamics show higher plume injection heights and larger variability. The changes in predicted fire emissions and injection height reduced the consistent over-predictions of $PM_{2.5}$ and $O_3$ seen in FireWork-Ops. On the other hand, there were minimal fire emission contributions to surface $NO_2$, and results from FireWork-CFFEPS do not degrade $NO_2$ forecast skill compared to the RAQDPS. Model performances statistics are slightly better for Canada than for the U.S., with lower errors and biases. The new system is still unable to capture the hourly variability of the observed values for $PM_{2.5}$, but it captured the observed hourly variability for $O_3$ concentration adequately. FireWork-CFFEPS also improves upon FireWork-Ops categorical scores for forecasting the occurrence of elevated air pollutant concentrations in terms of false alarm ratio (FAR), and critical success index (CSI).

Keywords: air quality modelling, air quality forecast, wildfire smoke, plume injection height

## 1. Introduction

With 28% of the world's boreal forest (552 million ha), and 9% of the world's forests, Canada experiences frequent wildland fires (Natural Resources Canada, 2018). These fires are an integral part of the forest lifecycle: they regulate pests, release essential nutrients and open the forest canopy to new growth. Key tree species such as jack and lodgepole pines require the heat from fires to rejuvenate seedlings and facilitate new growth (Block et al., 2016). However, from the 19[th] until the later part of the 20[th] century, active fire suppression campaigns across North America resulted in an accumulation of fuel and an increase in tree density and fuel continuity in forests, which, upon ignition, can result in large intense wildfires (Block et al., 2016). In the past decades, both Canada and the United States have experienced wildland fires with large areas burned, causing significant economic costs and loss of life. In Canada, the area burned by fires has significantly increased in this past decade compared to the last 55 years (Landis et al., 2018), and for the provinces of Alberta (AB) and British Columbia (BC), the costliest fire seasons with the most people displaced in the last few decades took place in 2016 and 2017, respectively (Abbott and Chapman, 2018; MNP LLP, 2017). Similarly, according to statistics from the U.S. National Interagency Fire Center, the six years with the largest annual fire burned area (>3 million ha. y$^{-1}$) since 1984 have occurred in the past decade (2006-2018); in California, the state experienced its ten costliest wildfires in the period between 1990 and 2018 (National Interagency Fire Center, 2018; Struzik, 2017). The trend may continue since studies suggest that with climate change, the occurrence of higher temperature, increased drought, and earlier spring melt may further increase the frequency, size, and duration of wildfires across North America (Liu et al., 2013; Wotton et al., 2017).

Large-scale wildfires can pose direct threats to life and property, but they also produce large amounts of smoke with significant quantities of atmospheric pollutants. Primary air pollutants such as fine particulate matter ($PM_{2.5}$), nitrogen oxides ($NO_x$), carbon monoxide (CO) and ammonia ($NH_3$) are released during combustion processes, and depending on the fire condition and fuel type, various volatile organic compounds (VOCs), trace metals, and hazardous air toxics are also emitted (Akagi et al., 2011; Fraser et al., 2018; Hatch et al., 2017; Urbanski, 2014; Wentworth et al., 2018). These pollutants not only fumigate source areas, but they can also be transported across regions, and even over thousands of kilometers across continents (Jaffe, 2004; Teakles et al., 2017). Primary air pollutants from wildfires can also undergo chemical reactions to produce secondary air pollutants such as $O_3$ and secondary $PM_{2.5}$ that further deteriorate air quality (Baker et al., 2016; Jaffe and Wigder, 2012; Larsen et al., 2018; Valerino et al., 2017). Recent studies have shown increases in population exposure to $PM_{2.5}$ pollutions across North America as a result of wildfire activities (Munoz-Alpizar et al., 2017; Rappold et al., 2017). Such exposure can lead to a wide range of health problems, including asthma and congestive heart failure, and can also significantly exacerbate pre-existing heart or lung conditions (Cascio, 2018; Finlay et al., 2012; Liu et al., 2014). Downwind communities impacted by wildfire smoke plumes have been associated with increases in physician visits, hospital admissions, and premature mortality, especially in vulnerable populations such as the elderly, infants, and those of low socioeconomic status (Johnston et al., 2012; Liu et al., 2014; Reid et al., 2016). Continuing into the 21[st] century, the air pollution hazard from wildfires is expected to become even more important amid the ongoing reductions of anthropogenic emissions in many countries (Knorr et al., 2017).

In Canada, Environment and Climate Change Canada (ECCC) is mandated to provide air quality (AQ) forecasts for populated communities across the country (see Table S1 in the Supplementary Material (SM) for a list of the acronyms used in this paper). Air quality forecasts are disseminated to the public as an Air Quality Health Index (AQHI) (Stieb et al., 2008). The AQHI, which was developed jointly with Health Canada, is a simple-to-understand threat level index with a scale from 1 to 10+. The location-specific index is calculated based on ambient surface concentrations of $O_3$, $PM_{2.5}$, and $NO_2$ in population centres. The index is effective in relaying AQ conditions during smog episodes and forest fire smoke events (Fish et al., 2017). Vulnerable populations can adjust their daily activity to forecast AQHI levels, and many provincial and local health service agencies in Canada have

established guidelines based on AQHI levels to suggest limiting outdoor activities and thus reducing smoke exposure when under the influence of wildfire smoke (BC Ministry of Environment, 2017; Calgary Airsheds Council, 2017; Yukon Health and Social Services, 2017).

ECCC supports the AQHI program through the development and application of operational numerical AQ forecast models. Since 2001, the operational Regional Air Quality Deterministic Prediction System (RAQDPS) has provided twice-daily, 48-hour forecasts of $O_3$, $PM_{2.5}$, and $NO_2$ for Canada and the U.S. The system focuses on urban air pollution risks from anthropogenic pollutant emissions and meteorological conditions favorable to smog formation. To account for wildfire-related smoke events, the FireWork system (Pavlovic et al., 2016b) for North America was deployed operationally in 2016 to augment the RAQDPS with near-real-time (NRT) biomass burning emissions. With FireWork, AQHI forecasts that include the fire contribution to $PM_{2.5}$ concentrations are disseminated by regional forecasters to first responders and public health decision-makers. The two AQ modelling systems together with local measurements were demonstrated to be an effective approach for public health application (Yuchi et al., 2016).

Other numerical AQ forecast systems similar to FireWork are available in North America, albeit based on different approaches. The experimental High Resolution Rapid Refresh-Smoke system operated by the U.S. National Oceanic and Atmospheric Administration (NOAA) provides 36-hour forecasts for the continental U.S. utilizing satellite-fire-radiative-power (FRP)-derived emissions and the WRF-Chem model without considering atmospheric chemistry (http://rapidrefresh.noaa.gov/hrrr/HRRRsmoke/: Ahmadov et al., 2017). The AIRPACT system operated by Washington State University provides daily 48-hour forecasts for the states of Washington, Oregon and Idaho using the WRF-CMAQ model (http://www.lar.wsu.edu/airpact: Herron-Thorpe et al., 2014). In addition, the National Weather Service (NWS) of NOAA has a 48-hour smoke forecast product that uses both the WRF-CMAQ model and the BlueSky-HYSPLIT trajectory system (https://airquality.weather.gov: Lee et al., 2017; Stajner et al., 2011). Other trajectory-based systems with NRT fire emissions include the U.S. BlueSky system run by the U.S. Forest Service (USFS) (https://www.airfire.org/data/bluesky-daily: Larkin et al., 2009), and a Canadian counterpart from BlueSky Canada operated by the University of British Columbia (http://firesmoke.ca: Schigas and Stull, 2013).

The predictive skills of chemical transport models (CTMs), especially those that account for wildfire activity as does FireWork, depend critically on the accuracy of the input emissions. In most regional CTMs, anthropogenic emissions are based on annual national emissions inventories processed with activity adjustments to be hourly, and gridded to the model domain with proxy spatial surrogates (Matthias et al., 2018). On the other hand, biomass burning emissions, because of their high spatial and temporal variability, require NRT updates of fire information, hourly processing, and allocation to domain grid cells representing the fire location. The quantification of biomass burning emissions and parameterization of emissions distribution within AQ modelling systems have been topics of recent research. Zhang et al. (2014) demonstrated the sensitivity of the WRF-Chem CTM's aerosol loading, atmospheric transport, and radiative feedback to emissions inputs from seven different fire emissions inventories. Garcia-Menendez et al. (2014) showed the high variability of modelled $PM_{2.5}$ as a function of spatiotemporal distributions of input fire emissions with the U.S. Environmental Protection Agency (EPA) CMAQ CTM. While there have been advances on global fire emissions quantification with top-down approaches using direct satellite products, there have been fewer developments with bottom-up, fire process-specific approaches (Anderson et al., 2015; de Groot et al., 2007; Larkin et al., 2014; Ottmar, 2013).

In addition to the overall quantification of biomass burning emissions, emission injection heights are another important parameter in simulating the transport of smoke in global- and regional-scale CTMs (Paugam et al., 2016). Fire emissions must be vertically distributed within model grid columns to account for the heights where fires release emissions into the atmosphere. This is a small-scale process that depends on the atmospheric environment and the intensity of vertical transport generated by factors such as fire

heat flux, fire size, entrainment, and turbulence. An initial injection height either within or above the planetary boundary layer (PBL) has profound consequences on the transport of smoke constituents, and, in turn, on surface $PM_{2.5}$ concentrations near the source region and further downwind. Large fires can trigger plume injection to heights above the PBL and into the stable free troposphere where emissions can be transported over long distances by stronger winds. Several methods have been developed to treat fire emission injection height in CTMs, ranging from simple empirical approaches (Achtemeier et al., 2011; Sofiev et al., 2012) to those considering microphysics and smoke entrainment (Freitas et al., 2007; Rio et al., 2010) to fully integrated systems that resolve fire plume dynamics internally (Kochanski et al., 2014; Mandel et al., 2014). For operational numerical forecast systems such as FireWork, where model run time is a critical consideration, the parameterization typically follows either the industrial smokestack plume-rise approach of Briggs (Briggs, 1965) or a simple assumption of uniform vertical distribution below the modelled mixing height, both of which may result in an underestimation of long-range transport as their estimates of the plume injection height are generally located below the PBL height (Mallia et al., 2018). Furthermore, different interpretations and implementations of plume-rise parameterizations within CTMs can result in differences in modelled plume injection heights for both facility stacks and fire sources. For example, recent model experiments using the CMAQ model showed that a modified Briggs parameterization with estimated fire buoyancy heat flux can adequately capture plume injection heights from wildfires and prescribed fires (Baker et al., 2018; Zhou et al., 2018).

Recently, the Canadian Forest Service (CFS) of Natural Resources Canada has developed the Canadian Forest Fire Emission Prediction System (CFFEPS) to improve the representation of biomass burning emissions for AQ model applications. CFFEPS improves the current fire emissions processing used by FireWork through additional process-specific considerations, including an updated North American fuel map, closer integration with forecast meteorology for treating fire behaviour, updated emission factors, and an efficient fire injection plume height parameterization based on a fire-energy thermodynamic approach (Anderson et al., 2011). The combined changes are aimed at improving the overall emission estimates from biomass burning and the spatiotemporal representation of biomass burning emissions for input to a CTM simulation. In this work, we have integrated ECCC's FireWork system with CFFEPS v2.03 (referred to as FireWork v2.0 or FireWork-CFFEPS) to improve NRT processing of biomass burning emissions. The integration is aimed at improving FireWork forecasts of surface-level $PM_{2.5}$, $O_3$, and $NO_2$ across the domain while still allowing for timely delivery of forecast products to regional forecasters and emergency first responders.

In this paper, we describe CFFEPS itself and how it is integrated with the FireWork system, and we then evaluate the changes in predictive skill of surface pollutant concentration forecasts important for regional AQHI. FireWork-CFFEPS system performance is also compared to that of the current operational FireWork system (referred to as FireWork v1.0 or FireWork-Ops). We evaluate the model forecast skill with standard model performance statistical metrics for one recent fire season in 2017 (July-September). In Section 2 we first describe the current FireWork modelling system (FireWork-Ops), the new CFFEPS, and its integration into the FireWork system (FireWork-CFFEPS). Section 3 provides details of the model experiment setup for forecast evaluation and presents the forecast comparison with the operational FireWork system. This is followed by a discussion in Section 4 and summary and conclusions in Section 5.

## 2. Description of Modelling Systems

### 2.1 RAQDPS and FireWork

The ECCC operational Regional Air Quality Deterministic Prediction System (RAQDPS) and the FireWork system with NRT biomass burning emissions were previously described in detail (Munoz-Alpizar et al., 2017; Pavlovic et al., 2016b, 2016a) so only

a summary is provided here. The RAQDPS is ECCC's operational regional AQ forecast system that provides short-term forecasts of surface-level concentrations of $PM_{2.5}$, $O_3$, and $NO_2$ for calculation of the AQHI. The core CTM in the RAQDPS is the on-line, one-way-coupled GEM-MACH (Global Environmental Multi-scale Modelling Air quality and Chemistry) model, where a detailed representation of atmospheric chemistry, including emissions, dispersion, and removal processes, is contained in a module

embedded within the Global Environmental Multiscale (GEM) numerical weather prediction (NWP) model. GEM is run by ECCC in its operational global and regional weather forecast systems (Charron et al., 2012; Côté et al., 1998). The RAQDPS uses a 10-km latitude-longitude grid covering North America (Figure 1). The regional version of GEM uses a 300-second integration time step while the chemistry module of GEM-MACH employs an integration time step of 900 seconds to reduce computational expense (although operator splitting is employed so some processes in the GEM-MACH chemistry module such as gas-phase chemistry

are solved using much smaller time steps). In this operational version of GEM-MACH, there is no feedback from the chemistry module to GEM, so that while meteorology affects chemistry, chemical fields do not influence meteorology.

The GEM-MACH chemistry is in effect a multi-phase, multi-pollutant CTM that considers the interactions of gas-, aqueous-, and particle-phase chemical components. The gas-phase chemistry mechanism is based on an updated version of the ADOM-2 mechanism with 42 species and 114 reactions (Lurmann and Stockwell, 1989). The aqueous-phase chemistry mechanism is based

on an updated version of the ADOM mechanism with 13 species and 25 reactions (Fung et al., 1991). PM chemical composition is represented with eight chemical components: sulfate, nitrate, ammonium, elemental carbon, primary organic matter, secondary organic matter, crustal material, and sea salt. The operational version of GEM-MACH assumes a simplified two-bin PM size distribution with fine-PM and coarse-PM aerodynamic diameter size bins of 0-2.5 µm and 2.5-10 µm, respectively. The treatment of aerosol-phase dynamics is based on the Canadian Aerosol Module (CAM) with detailed aerosol processes including nucleation,

condensation, coagulation, dry deposition, cloud scavenging and aerosol-cloud interactions (Gong et al., 2003), and inorganic aerosol thermodynamics, cloud processing, and secondary organic aerosol chemistry are also considered (Gong et al., 2015; Makar et al., 2003; Stroud et al., 2011). The version of GEM-MACH from the operational RAQDPS that was considered in this study (RAQDPS019) was v.2.2.0 (Moran et al., 2018).

Emission files used by the RAQDPS include emissions from both anthropogenic and biogenic sources. The anthropogenic

emission inventories that are considered are updated every few years. For the 2017 operational runs considered here, RAQDPS anthropogenic emission files were based on the 2010 Canadian national Air Pollutant Emissions Inventory (APEI), the 2011 U.S. National Emissions Inventory (NEI), and the 1999 Mexican NEI. These inventories were processed using the SMOKE emissions processing system to generate files of hourly, gridded, chemically-speciated emissions fields (Zhang et al., 2018). Biogenic emissions are calculated online in the RAQDPS based on the algorithm from BEIS version 3.09 with BELD3-format vegetation

land cover for Canada and the U.S. It is worth noting that the RAQDPS anthropogenic input emissions were updated in Sept. 2018 based on the 2013 Canadian APEI, a projected 2017 U.S. NEI, and the 2008 Mexican NEI (Moran et al., 2018). In order to understand the impact of using inventories for older base years, Table S2 compares 2010/2011 and 2017 inventory values for several western Canadian provinces and northwestern U.S. states. Over this period, $NO_x$ and VOC emissions decreased by 8% and 8% in western Canada and by 31% and 9% in the northwestern U.S., respectively, whereas $PM_{2.5}$ emissions increased by 11%

and 1%. The actual magnitudes of these differences are comparable to or smaller than the estimates of $NO_x$, VOC, and $PM_{2.5}$ emissions from North American wildfires given in Table 5

The current operational version of FireWork (FireWork-Ops) is identical to the RAQDPS except for the inclusion of NRT biomass burning emissions. The fire information used by FireWork is obtained in real time from the CFS operational Canadian Wildland Fire Information System (CWFIS) (http://cwfis.cfs.nrcan.gc.ca; Lee et al., 2002). CWFIS provides fire activity and fire behaviour

information based on initial NRT fire hotspot data from three satellite sensors: the Advanced Very High Resolution Radiometer

(AVHRR); the Moderate Resolution Imaging Spectroradiometer (MODIS); and the Visible Infrared Imaging Radiometer Suite (VIIRS) (Englefield et al., 2004; Quayle et al., 2003). The AVHRR hotspots are obtained from NOAA; MODIS from the National Aeronautics and Space Administration (NASA) and the USFS Geospatial Technology and Applications Center (GTAC); and VIIRS from NASA, the University of Maryland, and GTAC. Image pixels containing actively burning areas are mapped with each satellite pass approximately every 6 hours. Note, though, that fire monitoring using satellite-derived hotspots has several drawbacks, the most significant being the inability of the sensors to penetrate clouds and thick smoke plumes. Geostationary satellites such as GOES are not used in this process despite their higher temporal frequency because their coarse spatial resolution over Canada is not suited to this type of mapping.

At each fire hotspot location CWFIS assigns local noon-time meteorology (surface temperature, humidity, 10-m open wind speed, and cumulative rainfall from past 24 hours) from measurements or GEM model forecasts, and modelled fire characteristics based on the Canadian Forest Fire Behavior Prediction (FBP) System (Forestry Canada Fire Danger Group, 1992), including total fuel consumption for the associated fire and fuel types. Inputs to FBP include fuel type, elevation, slope steepness, slope direction, and outputs from the Canadian Forest Fire Weather Index (FWI) system, used to estimate fuel moisture (Lawson and Armitage, 2008). In FireWork-Ops, the area burned (per day) is assumed to be 38.5 hectares per hotspot (350 m burn radius). The estimated daily total fuel consumption for each hotspot is then combined with emission factors from the USFS Fire Emission Production Simulator (FEPS), a component of the BlueSky Modelling framework (Larkin et al., 2009), to calculate daily emissions of $PM_{2.5}$, $PM_{10}$, $NO_x$, VOC, CO, and $NH_3$ at each hotspot location. Next, the daily emissions are allocated to hour-specific emissions using a fixed diurnal temporal profile (Western Regional Air Partnership, 2005), and the $PM_{2.5}$, $PM_{10}$, $NO_x$, and VOC emissions are chemically speciated to RAQDPS-specific model species using the SMOKE processor and wildfire-specific PM, VOC, and $NO_x$ speciation profiles. The final hour- and RAQDPS-specific fire emissions are combined with the other anthropogenic point source emissions for input into the GEM-MACH model. To represent plume rise, the treatment of fire emissions within the GEM-MACH model follows the same approach as for anthropogenic point sources, with initial emission injection height following the Briggs plume-rise parameterization (Akingunola et al., 2018).

Operationally, the FireWork system is run twice per day at 00 and 12 UTC during the Canadian wildfire season from April 1 to October 31. The RAQDPS is run on the same schedule but throughout the year. FireWork and RAQDPS have the same continental domain, as depicted in Figure 1, and the same 772 by 642 latitude-longitude grid with 0.9 degree horizontal grid spacing (~10 km). Since FireWork is identical to the RAQDPS except for the inclusion of NRT biomass burning emissions, by subtracting the RAQDPS forecast $PM_{2.5}$ concentration field from the FireWork forecast $PM_{2.5}$ concentration field, fire-specific $PM_{2.5}$ concentration (fire-$PM_{2.5}$) can be obtained as forecast guidance. Note that fire-$PM_{2.5}$ includes both primary and secondary contributions from primary $PM_{2.5}$ emissions and gas-phase emissions of PM precursors.

Although the FireWork system has demonstrated improved forecast skill for $PM_{2.5}$ over RAQDPS during the wildfire season (Pavlovic et al., 2016b), there are assumptions made in the current operational version of FireWork that can be improved to better represent the dynamics of fire emissions. We now describe a new modelling approach for calculating NRT biomass-burning emissions for FireWork that addresses some of these limiting assumptions.

## 2.2 Canadian Forest Fire Emissions Prediction System

The Canadian Forest Fire Emissions Prediction System (CFFEPS) is a new model to predict smoke plume development for Canada. Currently, the system consists of a fire-growth model, a fire emissions model, and a thermodynamic-based model to predict the vertical penetration height of a smoke plume from fire energy. CFFEPS makes use of outputs from CWFIS, and incorporates the Canadian Forest Fire Danger Rating System (CFFDRS), including the Canadian Forest Fire Weather Index (FWI) System (Van

Wagner, 1987) and the FBP System (Forestry Canada Fire Danger Group, 1992) in order to allow for adjustments based on hourly forecast meteorological fields. The new model also follows techniques used in FEPS and CONSUME 3.0, both developed by the USFS (Anderson et al., 2004; Prichard et al., 2006).

The prediction of smoke emissions and the energy generated from wildland fires requires estimating the amount of forest fuel consumed by fire, which in turn involves estimating the mass of fuel consumed, which is a product of area burned and fuel consumed per unit area.

### 2.2.1 Area Burned

Fire growth is dependent on fuel type, fuel moisture, weather conditions, terrain, and fire suppression activities. While a number of fire-growth models exist, ranging from simple elliptical growth to more sophisticated models capturing spread over heterogeneous fuel and terrain, tests conducted during the current implementation of CFFEPS with FireWork indicate that for smoke emissions estimation, fire growth is best captured by assuming daily persistence; that is, if a fire burns a certain area on a given day, it will burn an equal area the next day. Future attempts may be conducted to incorporate such fire-growth models in CFFEPS.

Over the course of a day, fire growth rates vary as the temperature and wind speed typically decrease at night, while the relative humidity increases. The moisture content of the fuel in the litter layer on the forest floor varies correspondingly. Typically, the Fine Fuel Moisture Code (FFMC) reaches a peak with minimum fuel moisture at 5:00 p.m. LST (Van Wagner, 1987). This affects the rate of spread (ROS) of a fire, which, in turn, increases its intensity and area growth. The CFFEPS model provides two approaches to capture diurnal variations in fire growth. The first is a simple top-hat approach where the daily growth is spread evenly over a fixed period of time (9:00 a.m. to 9:00 p.m. LST). A second approach, which was applied in FireWork-CFFEPS, uses a weighting scheme following an average diurnal pattern of the rate of spread based on the FFMC, which is diurnally adjusted over time using the technique developed by Lawson et al. (1996).

### 2.2.2 Fire Emissions

CFFEPS calculates wildfire emissions following the bottom-up approach, where a measure of activity, in this case effective biomass burned, is multiplied by emission factors for different chemical species. The effective biomass burned is calculated as total fuel consumption multiplied by the burn area. In CFFEPS, total fuel consumption is calculated by the FBP system driven with hourly forecast meteorology. This includes crown fuel consumption (CFC), surface fuel consumption (SFC), and their sum, total fuel consumption (TFC), in units of kg of dry biomass $m^{-2}$ or tonnes $ha^{-1}$. CFC represents the mass of tree foliage burned per unit area of the forest canopy above the ground. SFC represents fuel consumption per unit area of the forest-floor biomass. The bulk density of the forest-floor biomass increases with depth, which has an impact on dominant combustion type, fire rate of burn, and timing of emissions. Table 1 shows the 14 fuel types considered in CFFEPS and the bulk densities used for each fuel type. These values are based on the original bulk densities documented in the FBP literature and are summarized in Anderson (2000). Following the technique used by CONSUME, CFFEPS divides combustion into three stages: flaming, smoldering, and residual combustion. The residual combustion stage follows the smoldering stage and is an incandescent form of combustion with little or no visible smoke. Table 2 summarizes the combustion phase allocation factors used by CFFEPS to allocate fuel consumption to these combustion stages. For CFC, canopy allocation factors of 94% flaming combustion and 6% smoldering combustion are applied to the crown fuel mass. For SFC, allocation factors are broken down into three ground layers: the litter, upper duff, and lower duff layers. To estimate the amount of fuel consumed within each forest-floor layer, the depth of burn is calculated, by burning off the litter layer first, then the upper duff layer, and finally the lower duff layer. The total depth of burn is determined from the SFC value, fuel type, and corresponding fuel bulk densities (Table 1). For most fuels, it is assumed that the first 1.2 cm

describe the litter (FFMC) layer, the upper duff is represented by the 1.2-7 cm depth, and the lower duff is represented by the layer below 7 cm (Van Wagner, 1987). Bulk densities below 8 cm are assumed to equal that of the 6-8 cm layer. For slash fuels, the slash allocation factors are applied against TFC as there is no crown fuel, while for surface-only grass fuels, grass allocation factors are applied against the grass fuel load, which is assumed to be completely consumed.

The three combustion stages are considered to have three burn durations. Flaming combustion is considered to occur within the first 15 minutes with all emissions from flaming consumption immediately released into the atmosphere. Smoldering and residual combustion last for several hours depending on available fuel load. Assuming a constant forest floor smoldering rate of 1 cm h$^{-1}$ (Huang and Rein, 2019), CFFEPS calculates the duration of these stages based on the calculated depth of burn. After flaming, the first half of this time is assumed to be smoldering combustion, and the second half is assumed to be residual combustion.

Table 3 lists eight species-specific emission factors used in FireWork-CFFEPS. While FireWork-Ops uses average emission factors from FEPS, updated emission factors were chosen for CFFEPS based on recent literature (Urbanski, 2014). Depending on the three stages of combustion, time series of emissions released to the atmosphere are created for each pollutant in accordance with the emission factors and the duration of the combustion stages. These emission factors are applied for all input fuel types in the current application, although CFFEPS is now designed to allow for fuel-specific values as found in recent measurements (Liu
et al., 2017).

### 2.2.3   Plume Rise

Plume rise used in CFFEPS is based on the thermodynamic plume model developed by Anderson et al. (2011). This model predicts the penetration height of a plume based on the amount of energy injected by the fire into the atmosphere and an environmental
lapse rate.

The energy released from a wildland fire can be determined using Byram's equation

$$I = H \, w \, r \, , \tag{1}$$

where $I$ is the intensity of a fire per unit length of fire front [typically measured as kW m$^{-1}$], $H$ is the heat of combustion [kJ kg$^{-1}$], $w$ is the weight of the fuel consumed per unit area [kg m$^{-2}$] and $r$ is the rate of spread [m s$^{-1}$ but normally measured as m min$^{-1}$].
The heat of combustion $H$ is a constant, typically 18,000 kJ kg$^{-1}$ for dry wood, and represents the total release of heat during both flaming and residual combustion (Byram, 1959). Following a similar format, the energy released by the fire, $Q_{fire}$ [kJ] becomes

$$Q_{fire} = H \, w \, A \, , \tag{2}$$

where $A$ [m$^2$] is the area burned.

During combustion, not all of the energy released by a forest fire enters the plume; instead, the fire's energy is partitioned such
that a portion of the energy is projected ahead of the fire to heat fuels to combustion temperatures, or into the ground beneath the fire, etc. Thus, to calculate the energy injected into the plume, the following energy balance for a wildland fire was devised for CFFEPS:

$$Q_{plume} = Q_{fire} - Q_{moisture} - Q_{fuel} - Q_{radiation} - Q_{surface} - Q_{incomplete} \, , \tag{3}$$

where $Q_{plume}$ is the energy injected into the plume, $Q_{fire}$ is the total energy of the fire, $Q_{moisture}$ is the energy lost to evaporate moisture in the fuel, $Q_{fuel}$ is the energy required to heat the fuel to the temperature of combustion, $Q_{radiation}$ is the energy lost radiatively into space away from any fuels and the plume, $Q_{surface}$ is the energy injected into the ground only to be released after plume development, and $Q_{incomplete}$ is the energy lost due to incomplete combustion. These values are solved for in CFFEPS based on input parameters from FWI and FBP. In general, approximately 10-20% of the total energy of the fire enters the plume.

The plume energy is injected into the atmosphere above the fire, modifying the plume's temperature profile to a dry adiabatic lapse rate. The energy required to modify the atmospheric column above the fire can be calculated as the integral

$$q = -c_p \oint T \, dln\theta \quad , \tag{4}$$

where $q$ is the energy per unit mass [J kg$^{-1}$], $c_p$ is the heat capacity of dry air [1005 J kg$^{-1}$ K$^{-1}$], $T$ is the air temperature [K], and $\theta$ is the potential temperature [K]. Temperatures and potential temperatures are provided by the environmental and dry lapse rates while $q$ is derived from $Q_{plume}$, over the mass of the plume, $M_{plume}$, where

$$\frac{M_{plume}}{A} = \frac{p_s - p_t}{g} \quad , \tag{5}$$

and $A$ is the ground-level area of the fire, $p_s$ and $p_t$ are the pressures at the surface and the top of the plume, respectively, and $g$ is the gravitational acceleration (9.8 m s$^{-2}$).

The CFFEPS model allows for entrainment of environmental air into the fire's plume, which is represented by an entrainment angle α. Essentially, the plume rises in a conical fashion expanding with height. Informal study of dozens of wildfire plume photographs suggests 12° as an average entrainment angle value. When considering entrainment, the modified volume of the plume depends on fire size. Entrainment has a more significant role in smaller fires than in larger fires. For a large fire, the proportional volume contributed to the plume from entrainment is minimal, while on a small fire, it is substantial. When considering entrainment, the mass of the plume can be calculated using the average of the plume top area at the location of the fire.

As a plume rises, the density of smoke diminishes, as does the density of air. In CFFEPS, it is assumed that the mixing ratio of smoke emissions to clear air is constant in the plume due to convective mixing. Given the total mass of smoke emissions ($M_{emissions}$) and the mass of the plume ($M_{plume}$), the smoke mixing ratio ($r_{smoke}$) can be calculated as

$$r_{smoke} = \frac{M_{emissions}}{M_{plume}} \quad , \tag{6}$$

Using the mixing ratio of smoke to clear air, the density of the smoke with height becomes a function of the density of air. This allows the vertical distribution of the smoke within the plume to be specified.

## 2.3 Integrating CFFEPS in FireWork (FireWork-CFFEPS)

The CFFEPS model is integrated into FireWork for NRT processing of biomass burning emissions. Methods and data sharing between FireWork and CWFIS have been enhanced to enable the new features contained in CFFEPS. The structure of the integration of CFFEPS with GEM-MACH (Figure 2) illustrates the flow of information. One key difference with FireWork-Ops is the replacement of FEPS and SMOKE by CFFEPS. Hotspot and meteorological information are collected by CWFIS and associated with forest fuels. Hourly meteorological forecasts for the hotspot location are then collected and passed to CFFEPS. Hourly fire activity, emissions, and plume rise parameters are calculated by CFFEPS and provided to GEM-MACH. By combining

the fire emissions with anthropogenic and biogenic emissions, GEM-MACH then simulates the atmospheric dispersion and chemistry of pollutants from all sources. Details about a number of changes needed to complete the integration of FireWork and CFFEPS follow.

### 2.3.1 Fire Detection and Mapping - Canadian Wildland Fire Information System

Operationally, CWFIS continues to provide NRT fire data during the Canadian fire season. The CFS Northern Forestry Centre in Edmonton, Alberta collects hotspots detected nationally from MODIS, NOAA/AVHRR, and VIIRS satellite imagery. In FireWork-CFFEPS, the actively burning area at the time of detection is assigned based on historical area burned and hotspot statistics for each province and fuel type. In Canada, provincial and territorial agencies provide annual data on area burned. Given knowledge of the number and locations of hotspots, an average fire size per hotspot can be calculated for each fuel type provincially

and territorially, with a recalibration performed every year: the 2017 values ranged from 7.52 ha per hotspot for O1 (grass) in BC to 43.88 ha per hotspot for coniferous fuels in Quebec.

For each hotspot, a fire-growth simulation environment is assembled. This includes the forest fuel type, and 1200h LST weather and fire weather conditions. Daily noon weather observations from over 2,500 stations in Canada are collected and used to produce fire-weather and fire-behaviour maps based on CFFDRS. Surface conditions are interpolated between stations using an inverse-

distance weighted approach; surface air temperature is cooled at a standard atmosphere rate of 6.5 ºC per 1000 m (to match local topography); and fire-weather values are recalculated accordingly. Because of the time required to collect and process noon observations across Canada's six time zones, forecasted weather is often used to reflect the current day's conditions. The CWFIS uses a 250 m-resolution fuel types map of Canada based on the National Forest Inventory (Beaudoin et al., 2018), North American Land Cover (Commission for Environmental Cooperation, 2017), and other datasets. Fuel types in the U.S. are determined from

the 13 Anderson Fire Behavior Fuel Model (Anderson, 1982) map obtained from LandFire (U.S. Geological Survey, 2016) and reclassified to CFFDRS fuel types by expert opinion.

### 2.3.2 Global Environmental Multiscale (GEM) Model

All of the information described to this point is collected and processed by CFS as part of CWFIS, and it is then provided to ECCC's Canadian Centre for Meteorological and Environmental Prediction for further processing. Once active-fire information

is received from CWFIS, 72-hour point forecasts are created for each hotspot using the 10-km regional version of the GEM weather forecast model. Forecasted values include surface conditions (temperature, humidity and wind speed) along with upper-air conditions (temperature and height) at specific pressure levels (850, 700, 500, 250 hPa). CFFEPS then uses the GEM forecasted hourly weather to calculate fire behaviour. This includes hourly values of FFMC, ROS (m min$^{-1}$), crown fraction burned (CFB, %), SFC (kg m$^{-2}$), TFC (kg m$^{-2}$), and head fire intensity (HFI, kW m$^{-1}$). Because of the nature of the persistence approach for area

burned (Section 2.2.1), the GEM model forecast does not drive the fire area growth, but forecast meteorological values are used in calculating fuel consumption at each fire location. Therefore, for the second day forecast, the procedure is repeated with the same daily area growth but with variable hourly fuel consumption based on forecasted meteorology.

### 2.3.3 Fire Growth, Fire Energy and Plume Injection Height

The next step in CFFEPS is the synchronization of fire characteristics with the detected hotspot. A detection time (in LST) is

determined for fire behaviour prediction purposes. The detection time is synchronized to the appropriate hour of the forecast (in UTC) and the diurnal growth of the fire is then calculated from the detection time, fire size, and hourly fire behaviour.

Once the hourly growth rate is established, CFFEPS calculates fuel consumption for each fuel type by depth of burn, and flaming, smoldering and residual times for fire energy and emissions. The fraction of energy released during the three combustion stages

is based on the allocation factors in Table 2. Fuel consumption from the current hour's fire growth is thus spread out over time. The flaming stage is assumed to occur in the first 15 minutes of combustion. Afterwards, a fire is assumed to burn into the forest floor at a rate of 1 cm h$^{-1}$. The total burn time is calculated knowing the depth of burn from the total surface fuel consumption and fuel bulk density. The first half of this time is assigned to the smoldering stage and the second half to the residual stage.

Energy values are then calculated over time using the hourly fire growth (i.e., the change in area burned from one hour to the next) and TFC. While growth is dictated by the persistence scheme used, hourly and daily changes in FWI values provide diurnal and daily changes to the TFC and thus to the energy released. Hourly energy values injected into the fire plume (see Equation 3) are next used to estimate hourly plume rise. Plume rise is calculated in one of two ways. The traditional approach, as described in Anderson et al. (2011), heats the air above a fire, adjusting the environmental lapse rate above the fire to a dry adiabat. The

environmental lapse rate used for the column above the fire is a single average value, though the choice of lapse rate will vary depending on the predicted plume height. For example, if the lapse rate from the surface to 850 hPa predicts a plume height of over 2000 m, then the lapse rate from the surface to 700 hPa will be used, but if that predicted plume is above 4000 m, then the lapse rate from the surface to 500 hPa will be used. A new alternative method that calculates plume rise using all measurements from the detailed upper-air profile and integrating the energy piecewise through the atmosphere is now included in CFFEPS

although it was not used in this study.

Given the fire area growth and the fuel consumption, total emissions over time are calculated. The estimated hourly plume injection height at each fire location is used directly in GEM-MACH to distribute fire emissions below the representative model layers. Fire emissions from all three stages of combustion are distributed below the injection height, through the model grid column based on the calculated smoke mixing ratio. A smoke mixing ratio ($r_{smoke}$) over time is then calculated based on the total emissions and the

plume height and mass of the plume.

### 2.3.4  Emitted Species

Similar to the method for fire energy, CFFEPS manages emissions per species by accounting for hourly TFC at each stage of combustion. Once the hourly TFC (kg m$^{-2}$ h$^{-1}$) is calculated, emissions per species per hotspot are calculated (g h$^{-1}$) with a user-input ancillary file containing species-specific emission factors. The emission factors of Table 3 expressed as grams of emitted

species per kilogram of combusted fuel (g kg$^{-1}$) for the three combustion stages are used in the implementation of CFFEPS in FireWork. Although emissions factors can also be dependent on fuel type, current input has one default emission factor applied to all fuel types Table 3.

Prior to input in GEM-MACH, hourly lumped emissions of non-methane hydrocarbons (NMHC), PM$_{2.5}$ and PM$_{10}$ are speciated to GEM-MACH's model mechanism species. Emission speciation profiles for flaming and smoldering of total organic gas from the

EPA's SPECIATEv4.5 database (Simon et al., 2010) are applied to the CFFEPS's flaming, and combined smoldering and residual emissions, respectively (see Tables S4 and S5). Finally, FireWork-CFFEPS incorporates the same set of biogenic and anthropogenic emission sources as the operational RAQDPS system to forecast overall atmospheric pollutant concentrations.

### 2.4    Differences between Current Operational FireWork-Ops and FireWork-CFFEPS

The current operational version of FireWork (FireWork-Ops) uses FEPS and SMOKE modules for fire emissions input into the

GEM-MACH forecasts. The new setup, FireWork-CFFEPS, presented here replaces those modules with the CFFEPS module for fire dynamics and emissions. The principal differences between FireWork-Ops and FireWork-CFFEPS are the following:

- FireWork-Ops uses static hotspot sizes of 38.5 hectares for all hotspots and fuel types; FireWork-CFFEPS uses yearly updated hotspot sizes categorized by fuel type and by province and territory.

- FireWork-Ops uses the hotspot size (38.5 ha) as the area burned on the first day; FireWork-CFFEPS uses reverse growth from the detection time and fire size to create fire sizes for the initial hours of the forecast.

- FireWork-Ops uses TFC as the flaming consumption, and the difference between Forest Floor Fuel Consumption (de Groot et al., 2009) and TFC as the smoldering consumption; FireWork-CFFEPS uses TFC, breaking it down into flaming, smoldering and residual combustion stages by fuel type and depth of burn (Table 2).

- FireWork-Ops applies a fixed diurnal profile for hourly allocation of the combined flaming and smoldering emissions from daily total fire emissions; FireWork-CFFEPS allots 15 minutes for flaming, and establishes the remaining period of burn using the depth of burn and an assumed burn rate of 1 cm h$^{-1}$. The latter period is divided equally between smoldering and residual combustion.

- FireWork-Ops does not consider fire energy; FireWork-CFFEPS calculates fire energy over time and uses that value to calculate plume injection height.

- FireWork-Ops uses the Briggs plume-rise parametrization with fixed plume temperature and plume velocity; FireWork-CFFEPS uses the fire energy thermodynamics approach to estimate hourly plume injection height and to distribute smoke based on smoke mixing ratio and air density.

- FireWork-Ops uses fixed emissions factors predefined by FEPS for seven species (PM$_{2.5}$, PM$_{10}$, CO, NH$_3$, NO$_x$, SO$_2$, NMHC); FireWork-CFFEPS has user-defined emissions factors that are dependent on combustion stages and fuel type.

- FireWork-Ops allocates lumped NMHC fire emissions to GEM-MACH model VOC species following a default profile; FireWork-CFFEPS allocates lumped NMHC fire emissions using separate flaming and smoldering speciation profiles.

## 3. FireWork-CFFEPS Forecast Experiment Evaluation

To assess the forecast performance of FireWork-CFFEPS, the system was run in hindcast mode for 2017, a recent year with high fire activity, and results were compared against the forecast performance of the operational FireWork-Ops system. Model forecast performance was assessed by comparing simulation results with available hourly, continuous surface measurements from the Canadian National Air Pollution Surveillance (NAPS http://maps-cartes.ec.gc.ca/rnspa-naps/data.aspx) network and the U.S. EPA Air Quality System (AQS https://www.epa.gov/aqs) for the three species important to the AQHI calculations, namely PM$_{2.5}$, O$_3$, and NO$_2$. Figure 3 shows the spatial distribution and number of the measurement stations that were considered. For both networks, measurement stations with less than 75% measurement completeness were removed to ensure temporal representativeness. Model hourly results for the near-surface concentrations at measurement site locations were extracted, paired by time, and evaluated using common model evaluation statistics as well as three operational, forecast-oriented categorical scores (Jolliffe and Stephenson, 2012). The categorical scores, calculated from hourly values, were probability of detection (POD), false alarm ratio (FAR), and critical success index (CSI), where

$$POD = \frac{Hit}{Observed} \ , \tag{7}$$

$$FAR = \frac{False\ Alarm}{Forecasted} \ , \tag{8}$$

$$CSI = \frac{Hit}{Hit + False\ Alarm + Miss} \ , \tag{9}$$

These three metrics quantify the model's skill in forecasting extreme events based on exceedance of threshold values and provide key guidance for operational forecasters in issuing AQ bulletins. The threshold values used for $PM_{2.5}$, $O_3$ and $NO_2$ were 30 µg m$^{-3}$, 65 ppbv, and 30 ppbv, respectively. These are thresholds tailored according to potential AQHI calculation across the region, and for $PM_{2.5}$ and $O_3$, they also agree with the Canadian ambient AQ standards for 24-hour $PM_{2.5}$ and 8-hour $O_3$ concentrations. They are calculated by counting the number of forecasted and measured data pairs that fall in the four binary categories shown in Table 4.

## 3.1    2017 Fire Season Model Forecast Evaluation

2017 was a significant fire year in both Canada and the U.S. with record fire starts and burned areas mostly in western states and provinces. In Canada, the total burned area for the 2017 season is shown in Figure 4 against a 10-year average. Although the fire season started slowly in May and June, with numbers below the 10-year average, fire activity then picked up very rapidly in July with several large fires in BC. Fire starts continued in August with fires in the Northwest Territories (NT), northern Alberta (AB), northern Saskatchewan (SK), central Manitoba (MB), and western Ontario (ON). Wildfires in western Canada were active until early September, with most fires occurring in south-central BC during July and August. Across Canada, BC had the highest number of fire hotspots, accounting for more than 50% of the Canadian total. Due to the severity of the wildfires in BC, the province declared a State of Emergency from July 7 until September 15. More than 1.2 million hectares were burned and more than 65,000 people were evacuated during this period. The Plateau Complex fire in south central BC was the single largest fire in the province on record, with a combined total fire area of 545,151 ha (Abbott and Chapman, 2018).

In the U.S., 2017 was the one of the most expensive years on record with respect to total firefighting costs with total federal spending close to $3 billion dollars (National Interagency Fire Center, 2018). Total burned area nationally was reported to be more than 4 million ha from 71,000 fires, significantly higher than the 10-year average of about 2.7 million ha. For states near Canada, fire activity was significant from August through mid-September for Washington (WA), northern Idaho (ID), western Oregon (OR), and western Montana (MT). Most notable was the Lodgepole complex fire in MT that burned 110,000 ha. It was the largest fire in MT history and also the largest in the U.S. for the 2017 season. The Chetco Bar fire in OR started in mid-July and burned 77,000 ha while the Diamond Creek fire in WA burned 52,000 ha. As a result of smoke plumes from these local fires and smoke plumes from BC wildfires, several cities in WA, OR, ID issued AQ advisories, with the air quality index reaching the highest, "hazardous", level.

### 3.1.1    Fire Emissions Comparison

FireWork-CFFEPS was run with the same model setup as the operational FireWork-Ops for the July-September 2017 period. Figure 5 shows the monthly total effective biomass burned from FireWork-CFFEPS and FireWork-Ops for hotspots greater than 5,000 tonnes burned per month. The spatial distributions of fire locations as clusters of hotspots for the two systems were generally similar, which was expected given that the same fire information was provided by CWFIS in both cases. However, the effective-biomass-burned total and the number of fires above the 5,000 tonnes threshold were different. The largest driver of the difference is the estimated burn area, which changed from a constant 38.5 ha per hotspot in FireWork-Ops to varying burn areas by province and fuel type in FireWork-CFFEPS. In BC, an overall reduction in burn area ranging from 7.5 ha per hotspot for grass fuels (O1) to 14.5 ha per hotspot for boreal mixwood (M1) greatly reduced the effective biomass burned across the province for all three months. Similarly, for SK, the increases in effective biomass burned can be attributed, in part, to increases in estimated burn area to approximately 40 ha per hotspot for boreal spruce and pine fuels (C2-C4).

In addition, the number of hotspots produced by fires above the 5,000 tonnes threshold are different, especially in August with more hotspots in FireWork-CFFEPS than FireWork-Ops for areas of northern AB, SK, MB and western ON. The changes are due to the combination of changes in estimated burn area and changes in fuel consumption driven by hourly forecast meteorology in CFFEPS. Variations in hourly meteorology can change the diurnal variation in biomass burned in CFFEPS, whereas in FireWork-Ops, the total effective biomass burned is calculated from daily totals based on local noon-time meteorology at each hotspot location.

Emissions totals were also quite different between the two systems as a result of the combined changes in effective biomass burned and in the process-dependent species emission factors. Table 5 summarizes the emission totals for the same three months from FireWork-Ops and the percentage difference for FireWork-CFFEPS by species and by country, as well as for individual provinces in Canada and U.S. states near Canada that were selected for their high fire activities. At the continental scale, FireWork-CFFEPS has consistently lower emissions than FireWork-Ops for VOC, $NO_x$, and $NH_3$ (-33% to -47%), yet significantly higher emissions for $PM_{2.5}$ and $PM_{10}$ (87%–88%). There are also small increases in CO emissions (<+10%), and small decreases in $SO_2$ emissions for fires in Canada (-6%) and the U.S. (-14%). At the regional scale however, the changes in emissions are very different as a result of (a) the fuel-type-dependent fire areas, (b) meteorology-influenced fuel consumption. and (c) dominant fuel types within each province and state. For example, in BC, where the dominant fuel types (Ponderosa pine and Douglas fir) have shallower fuel beds, there are large reductions (-36% to -75%) for VOC, $NO_x$, CO, and $SO_2$ emissions with FireWork-CFFEPS and comparatively small increases (7%) in $PM_{2.5}$ and $PM_{10}$ emissions. On the other hand, for AB, SK, and MB, the three prairie provinces east of the Rocky Mountains, fuel types are primarily boreal spruce and mixed-wood with deeper fuel beds, and there were large systematic increases (13% to 40%) in all emissions species with FireWork-CFFEPS compared to FireWork-Ops.

### 3.1.2    Plume Injection Height Comparison

Fire plume injection heights are calculated hourly at each hotspot location in both FireWork-Ops and FireWork-CFFEPS but by different methods, and fire emissions are distributed vertically within the model grid column below the modelled plume injection heights. FireWork-Ops parameterizes the plume injection height based on the Briggs parametrization, similar to those used in anthropogenic point sources specific to facility stacks. FireWork-CFFEPS applies the new fire-energy thermodynamic balance approach with forecasted hourly environment lapse rate at hotspot locations (see Section 2.2.3).

Figure 6 shows the injection-height frequency distribution by 200-meter altitude bin for all BC fire hotspots in August 2017 grouped by forecast hour as predicted by FireWork-CFFEPS and FireWork-Ops 48-hour 00 UTC forecasts. The frequency distributions for both FireWork-Ops and FireWork-CFFEPS display clear diurnal variability in modelled injection height throughout the forecast period with generally higher injection heights during local daytime (f00, f03, f18, f21). There are also large differences, however, in the distribution of injection heights between the two systems for the same hour, with FireWork-CFFEPS typically showing wider distributions and higher modelled injection heights than FireWork-Ops. During local daytime, the FireWork-CFFEPS injection-height frequency distribution ranged mostly from 2 to 6 km with its mode at around 4 km, whereas FireWork-Ops has a narrower distribution ranging from 1 to 3 km with its mode at around 1.5 km. The highest plume injection heights for FireWork-CFFEPS reach as high as 6 km, whereas FireWork-Ops modelled injection heights are always below 4 km under the same conditions. During local nighttime (f06, f09, f12, f15), the injection-height distribution for FireWork-CFFEPS ranges from 1 to 4 km with its mode at either 1.8 km or 3.2 km, whereas the injection-height distribution for FireWork-Ops is consistently below 2 km, with most hours having 50% or more injection heights below 200 m.

The large differences in modelled plume injection heights between FireWork-Ops and FireWork-CFFEPS result from the CFFEPS parameterization considering fire growth, fire energy, and the forecast environmental lapse rate. FireWork-Ops parameterizes

plume injection height based on a prescribed constant fire emission temperature, initial height, and modelled hourly PBL following the Briggs parameterization. In a recent study on model plume-rise parameterization, Akingunola et al. (2018) demonstrated that the current implementation of Briggs in GEM-MACH under-predicts measurements from facility stacks and can be further improved with a layered lapse-rate approach that is not currently used in the RAQDPS. A separate analysis also showed that FireWork-Ops injection height is limited to hourly PBL height with maximum injection height always equal to or less than the PBL height. This confines the vertical distribution of fire emission to near the Earth's surface and limits the amount of emissions penetrating into the free troposphere, where stronger winds enhance long-range transport. A detailed verification comparing CFFEPS-derived fire plume injection heights with surface observations and satellite-based estimates is beyond the scope of this paper. Nevertheless, the altitude range of FireWork-CFFEPS is in general agreement with a recent global fire-plume injection-height analysis from satellite remote sensing for the region (Val Martin et al., 2018), and the injection heights calculated by CFFEPS are not restricted to below PBL height as in FireWork-Ops.

### 3.1.3    Continental-scale Model Forecast Evaluation

The contributions of biomass burning emissions to modelled $PM_{2.5}$ concentrations can be obtained by subtracting the $PM_{2.5}$ concentrations of the operational RAQDPS model outputs from the FireWork outputs. Figure 7 compares mean monthly surface fire-$PM_{2.5}$ concentration for July–Sept. 2017 for FireWork-CFFEPS vs. FireWork-Ops. As expected, the spatial distribution of fire-$PM_{2.5}$ is closely related to the location of fire hotspots, with higher concentrations predicted near source areas. The spatial impact of fire-$PM_{2.5}$ over the continent can also be large due to transport of emissions downwind as demonstrated in earlier studies (Munoz-Alpizar et al., 2017; Rappold et al., 2017).

The overall spatial extent of fire-$PM_{2.5}$, however, is slightly different between FireWork-CFFEPS and FireWork-Ops. For all three months the mean forecasted near-source fire-$PM_{2.5}$ concentrations are generally lower for FireWork-CFFEPS than for FireWork-Ops, and the spatial extent of high fire-$PM_{2.5}$ concentrations (>40 µg m$^{-3}$) is also smaller. On the other hand, the spatial extent of fire-$PM_{2.5}$ for lower concentrations (<20 µg m$^{-3}$) is larger for FireWork-CFFEPS than for FireWork-Ops despite identical forecast meteorology. This larger spatial influence at lower concentration levels can mainly be attributed to the revised plume-rise parameterization approach in CFFEPS. The Briggs' parameterization in FireWork-Ops limits the vertical distribution of emissions to the PBL, thus reducing long-range transport of emissions, whereas the new parametrization from CFFEPS, with generally higher modelled plume injection heights, lofts some fire emissions above the PBL, where they can be transported longer distances by stronger winds in the free troposphere.

Model forecast performance for $PM_{2.5}$, $O_3$ and $NO_2$ has been evaluated for four geographic areas within the domain (Figure 1) for July-Sept. 2017. Daily maximum values for the three pollutants were paired against surface measurements (Figure 3) and grouped by station within these regions. Table 6 summarizes the model forecast performance statistics for the two FireWork systems and the RAQDPS for five basic statistics: observed mean ($\bar{O}$), modelled mean ($\bar{M}$), mean bias (MB), Pearson correlation coefficient (R), and root-mean-square error (RMSE). Due to comparatively lower fire activity in the eastern part of the domain in 2017, performance statistics for eastern Canada (ECAN) and the eastern U.S. (EUSA) show little difference across the three systems, and the modelled averages for $PM_{2.5}$, $O_3$ and $NO_2$ from FireWork-Ops and FireWork-CFFEPS are very similar to those of the RAQDPS. It is also evident from the MB scores that, like the RAQDPS, both FireWork systems over-predict concentrations for $O_3$ and $NO_2$, likely associated at least in part with uncertainties in the anthropogenic emissions.

For western Canada (WCAN) and the western U.S. (WUSA), Table 6 shows much larger differences between the three model versions due to the influence of fire activities in the area. Since western wildfires were a large contributor to $PM_{2.5}$ concentrations in 2017, observed $PM_{2.5}$ concentrations were much higher in these two regions with mean daily maxima of 23 µg m$^{-3}$ for both

WCAN and WUSA. The RAQDPS, with no contribution from fire emissions, shows significant under-predictions, with means of 12 µg m$^{-3}$ and 15 µg m$^{-3}$ for WCAN and WUSA, respectively, and low R values. With the inclusion of NRT biomass burning emissions in the two FireWork systems, model forecasts of PM$_{2.5}$ improve, with FireWork-CFFEPS showing consistently better performances. FireWork-Ops over-predicts, with mean forecast values of 44 µg m$^{-3}$ for both WCAN and WUSA, whereas FireWork-CFFEPS, while still over-predicting, has lower mean forecast values of 29 µg m$^{-3}$ and 31 µg m$^{-3}$, respectively. RMSE and R statistics also show systematic improvements for both regions with the CFFEPS emissions. Similar trends were also observed for O$_3$ and NO$_2$, with FireWork-CFFEPS outperforming FireWork-Ops, albeit with slightly poorer performance for O$_3$ compared to the RAQDPS but slightly better performance for NO$_2$. The improved performance of FireWork-CFFEPS compared to FireWork-Ops can be directly attributed to the revised biomass burning emissions and plume injection heights calculated by CFFEPS.

The categorical score comparisons for the three modelling systems for PM$_{2.5}$, O$_3$, and NO$_2$ for the two western regions are shown in Table 7. For PM$_{2.5}$, both FireWork systems again show large improvements over the RAQDPS for POD, FAR, and CSI, but the improvements are smaller for O$_3$ and absent for NO$_2$. The categorical scores for Firework-CFFEPS are better than FireWork-Ops for WCAN, with lower FAR and higher CSI. Although FireWork-Ops has a higher POD score, this is due to the system's over-predictions and is an inherent weakness of this particular score. Note that for the ECAN and EUSA regions, with low fire activity, the inclusion of biomass burning emissions in both of the FireWork systems has little impact on forecasts compared to the RAQDPS, and all systems have nearly identical categorical scores (not shown).

### 3.1.4 Model Forecast Evaluation for Wildfire Regions

Wildfire activity was most severe in August and early September 2017, especially between August 6 to 19 (weeks 15-16 in Figure 4) with a record burned area of more than 1.1 million hectares across Canada. Although most of the fires occurred in central BC, there was also significant fire activity in WA, ID, OR, and MT (Figure 7), which caused widespread PM$_{2.5}$ pollution across the region with several measurement stations recording hourly PM$_{2.5}$ concentrations in excess of 200 µg m$^{-3}$. The mean August PM$_{2.5}$ concentration from 79 NAPS measurement stations in AB and BC (AB+BC region) was 31 µg m$^{-3}$, and it was 34 µg m$^{-3}$ from 89 AQS measurement stations in ID, MT, OR and WA (ID+MT+OR+WA region). Both regions are shown in Figure 1. Wildfire activity was also observed across southern NT, northern AB, MB, and SK, and western ON (Northern-Canada region), which resulted in elevated PM$_{2.5}$ conditions in otherwise pristine regions of northern Canada. For the Northern-Canada region, due to the sparsity of NAPS stations, 10 measurement stations spread across NT, MB, SK, and western ON were chosen to represent the forecast conditions for the region (see Figure 1 and Table S3 for station locations). In this section, we focus on the August to early September period for these three regions to assess the day-to-day regional forecast performance of the FireWork-CFFEPS, FireWork-Ops, and RAQDPS systems during a period dominated by wildfires.

Figure 8 shows mean daily maximum PM$_{2.5}$ concentration time series from Aug. 1 to Sept. 18, 2017 for the three forecast simulations and the corresponding measurements averaged across the monitoring stations in each of the three regions. The periods of high pollutant concentrations as a result of wildfire activity can be identified as days with high observed PM$_{2.5}$ concentrations. For AB+BC and ID+MT+OR+WA regions, the weeks of August 1, 7, 28 and September 4 have mean daily maximum PM$_{2.5}$ concentrations close to or exceeding 50 µg m$^{-3}$. Throughout these periods, both FireWork systems over-predicted surface PM$_{2.5}$ concentrations, but with FireWork-Ops showing consistently higher positive bias than FireWork-CFFEPS. The large over-prediction of PM$_{2.5}$ by FireWork-Ops, despite having lower regional fire emissions compared to FireWork-CFFEPS, can be attributed to its lower modelled plume injection heights that trap emissions closer to the surface. Similar time series plots from individual stations (not shown) indicate that the over-predictions are consistently higher for stations closer to locations of fire

hotspots. During the weeks of lower regional surface PM$_{2.5}$ concentrations in August 15, 21 and September 11 as a result of lower fire activity, both systems captured the measured concentration trend, with FireWork-CFFEPS showing slightly better forecast performance than FireWork-Ops.

The model performance statistics for both western regions (Table 8) show systematic improvement of FireWork-CFFEPS over FireWork-Ops, with higher R and lower RMSE and MB scores. The measured mean daily maximum PM$_{2.5}$ concentration for the period for the AB+BC and ID+MT+OR+WA regions were 28 μg m$^{-3}$ and 42 μg m$^{-3}$, respectively; corresponding FireWork-Ops predicted mean values of 66 μg m$^{-3}$ and 126 μg m$^{-3}$, respectively, were large over-predictions, whereas FireWork-CFFEPS predicted mean values of 40 μg m$^{-3}$ and 71 μg m$^{-3}$ were considerably closer to the measurements. Between the two regions, the model forecast statistics were generally better for the stations in the two Canadian provinces than those for the four U.S. states. The PM$_{2.5}$ categorical scores for a threshold above 30 μg m$^{-3}$ (Table 9) also showed better forecast skill for FireWork-CFFEPS in terms of lower FAR and a slight increase in CSI.

For the Northern-Canada region, due to the sparsity of measurement stations over this large area and to stations being situated further away from fire hotspots and from sources of anthropogenic emissions, mean daily maximum surface PM$_{2.5}$ concentrations are lower with a measured mean daily maximum concentration of 20 μg m$^{-3}$ (Table 8). There were several days during the period when mean daily maximum surface PM$_{2.5}$ concentrations exceeded 25 μg m$^{-3}$ (Figure 8). The timing of high surface PM$_{2.5}$ concentration during the Aug. 6-19 period, which correlates closely with fire activity, is indicative of wildfire smoke influences. The forecast concentration time series from FireWork-CFFEPS and FireWork-Ops are more similar in this region, with both models closely matching the observed averages and displaying considerably better forecast skill than the RAQDPS. It is worthwhile to point out that unlike the two western regions described earlier, measurement stations in the Northern-Canada region were located further away from fire hotspots, and FireWork-Ops under-predicted observed mean daily maximum PM$_{2.5}$ concentration with a forecast average of 17 μg m$^{-3}$ (MB=-3 μg m$^{-3}$) while FireWork-CFFEPS did better with a forecast PM$_{2.5}$ average of 19 μg m$^{-3}$ (MB=-1 μg m$^{-3}$). Again, this improvement is due to changes in the plume injection height parameterization in FireWork-CFFEPS that allow for fire-PM$_{2.5}$ to be transported longer distances, resulting in higher surface concentrations further downwind. The PM$_{2.5}$ categorical scores for this region, however, showed only small differences between the two FireWork systems due to infrequent exceedances over the 30 μg m$^{-3}$ threshold. In contrast, the categorical score for the RAQDPS showed no skill for predicting wildfire smoke events, with 0% for POD and CSI because the model forecast PM$_{2.5}$ concentrations without NRT fire emissions never exceeded the event threshold throughout the analysis period.

Model performance by forecast hour was also analysed for these regions to compare the diurnal variability predicted by the three AQ forecast systems. Surface PM$_{2.5}$ concentrations from Aug. 1 to Sept. 16 were averaged by forecast hour (f00-f48) over all stations within the region and paired against similar values from measurements. Since FireWork-Ops is run twice a day with 00 UTC and 12 UTC starting times, these morning and evening runs were treated as different simulations and analysed separately. Figure 9 shows the hourly forecast comparisons from the three systems and the corresponding measured concentrations separated by model initialization hours for the three regions. In both the AB+BC and ID+MT+OR+WA regions, the magnitudes of hourly forecast concentration are better captured by FireWork-CFFEPS, which over-predicted the night-time concentrations but captured the lower day-time concentrations, whereas FireWork-Ops showed consistent over-prediction and the RAQDPS showed consistent under-prediction throughout the 48-hour forecasts. Interestingly, the predicted diurnal concentration variability is higher for all three forecast systems than in the corresponding measurements, with higher concentrations predicted at night and lower concentrations predicted during the day. For the Northern-Canada region, where the observed mean hourly concentration was the lowest at around 10 μg m$^{-3}$, both FireWork systems show consistent under-prediction but with FireWork-CFFEPS predicting slightly higher forecast surface PM$_{2.5}$ concentrations, consistent with the enhanced long-range transport from western fire locations

described earlier. The hourly PM$_{2.5}$ concentration trends are very similar between the 00 UTC and 12 UTC forecasts for all regions, except for the 12 UTC run in for the Northern-Canada region, where the FireWork-CFFEPS shows similar concentration range as FireWork-Ops for a few hours for the second-day forecasts.

It is evident from Figure 9 that for AB+BC and ID+MT+OR+WA regions, where the measurement stations were located closer to fire hotspots, both FireWork systems showed larger diurnal concentration variability than the corresponding surface measurements. The same diurnal variability, albeit with lower magnitude, is present in the diurnal curves for the RAQDPS without fire emissions. This indicates that the diurnal concentration variability over these regions must be determined largely by meteorology and less so by the treatment of the hourly emissions distribution, since the latter is different in FireWork-CFFEPS and FireWork-Ops.

Similar analyses were carried out for predicted daily maximum surface O$_3$ and NO$_2$ with available measurements for the three regions for the same high-wildfire analysis period. The results, presented in SM Tables S6 to S9 and Figures S1 to S4, show that FireWork-CFFEPS improved O$_3$ forecasts for all three regions compared to FireWork-Ops, with consistently better model performance statistics and categorical scores. The improvements were achieved by reducing the over-predictions seen in FireWork-Ops during fire events due to decreases in precursor emissions (Table 5). On the other hand, there were minimal changes to the model forecast performance for NO$_2$ due to smaller relative contributions from fire emissions and the shorter atmospheric lifetime of NO$_2$ relative to O$_3$. Similar to the domain-wide model evaluation (Table 6 and Table 7), FireWork-CFFEPS did not degrade the NO$_2$ model forecast performance of RAQDPS (Table S7 and Table S8).

Comparing Figure 9 with Figures S3 and S4 it is evident that there is stronger diurnal variability in the hourly concentrations of surface O$_3$ and NO$_2$ than PM$_{2.5}$ due to the daytime photolysis-driven reactions of O$_3$ and NO$_2$. The forecast models all captured this variability adequately but exaggerated it in the case of NO$_2$. Hourly concentrations of O$_3$ predicted by FireWork-CFFEPS are slightly higher than for the RAQDPS, showing the impact of increased emissions of O$_3$ precursors from fires. Of particular note, though, is that FireWork-CFFEPS corrects the consistent O$_3$ over-predictions in FireWork-Ops, although O$_3$ daily maximum concentrations are still slightly over-predicted, especially for the Northern-Canada region. Low O$_3$ concentrations at night are accurately simulated for all three regions. For NO$_2$, FireWork-CFFEPS hourly forecasts are similar to those of the RAQDPS, whereas FireWork-Ops forecasts higher early morning concentrations for both the AB+BC and ID+MT+OR+WA regions. Similar to the RAQDPS, FireWork-CFFEPS over-predicted high concentrations of NO$_2$ at night and under-predicted low concentrations during the day for the two Canadian regions. Note, though, that for the ID+MT+OR+WA region the measured NO$_2$ concentrations were available from only one AQS station in Portland, OR. All three forecast systems show consistent over-prediction of hourly NO$_2$ concentrations for Portland, but with FireWork-Ops predicting even higher early morning NO$_2$ concentrations than either FireWork-CFFEPS or the RAQDPS.

One additional evaluation was conducted for the Aug. 1 to Sept. 18, 2017 period to examine model performance for just those stations and days observed to be affected by wildfire plumes. Table S10 presents performance statistics for model predictions of daily maximum PM$_{2.5}$, O$_3$, and NO$_2$ concentrations for a filtered subset of measurement for which observed daily maximum PM$_{2.5}$ levels at individual stations were above 50 µg m$^{-3}$. Table S10 can be compared with Tables 8, S6, and S8, but it includes only 22% of the daily maximum PM$_{2.5}$ measurements, 14% of the daily maximum O$_3$ measurements, and 15% of the daily maximum NO$_2$ measurements considered in those three tables. Although both observed and modelled values are higher in Table S10 than the other three tables, the ranking of relative model performance is in general agreement with the analyses that considered all measurement stations within the regions and all days in the evaluation period.

### 3.1.5 Comparison of PM₂.₅ Vertical Column Density with Satellite Imagery

$PM_{2.5}$ vertical column density (VCD; g m$^{-2}$) is calculated as the modelled $PM_{2.5}$ concentration at a given layer multiplied by layer thickness and summed over the model grid column. Animations of fire-$PM_{2.5}$ VCD over North America are part of the product suite available from the operational FireWork system. Visual comparisons of forecast fire-$PM_{2.5}$ VCD with available satellite true colour images are regularly conducted as part of ongoing subjective evaluation of model performance. Figure 10 shows one such comparison for the two FireWork systems versus NASA's VIIRS true colour image for Aug. 14, 2017. The FireWork forecasts are from the previous day run valid for the same time at 12 UTC. The hotspots used to calculate wildfire emissions are overlaid on the model output as red dots.

The satellite image shows a large cluster of fires in NT, just north of the border with AB and SK, as well as other fires burning across northern AB, SK, and MB and in central BC. The dense smoke plume, influenced by the upper-level jet stream, blankets the entire NT region with tendrils extending southward along the eastern borders of AB and SK. Comparison of the next-day satellite image with the one-day forecasts of fire-$PM_{2.5}$ VCD shows good representations from the FireWork systems both in terms of hotspot locations and the spatial extent and distribution of the wildfire smoke. However, despite the spatial similarities, there are subtle differences in the fire-$PM_{2.5}$ VCD results between FireWork-CFFEPS and FireWork-Ops. First, the magnitude of fire-$PM_{2.5}$ VCD from FireWork-CFFEPS is higher in NT, especially just north of the fire hotspots, with values up to 0.05 g m$^{-2}$ compared to FireWork-Ops at 0.01 g m$^{-2}$. Second, the distribution of the mid-level contour (0.01 – 0.03 g m$^{-2}$) is less widespread in FireWork-CFFEPS, whereas the distributions of lower-level contours (<0.01 g m$^{-2}$) are more extensive in FireWork-CFFEPS than FireWork-Ops. Lastly, despite the lower surface $PM_{2.5}$ concentrations predicted by FireWork-CFFEPS for the AB+BC region (Table 8), the FireWork-Ops fire-$PM_{2.5}$ VCD field shows low to negligible contribution for central BC compared to much higher values in the FireWork-CFFEPS fire-$PM_{2.5}$ VCD field just southeast of fire hotspots. These differences can be attributed to the model sensitivity to the vertical emission distribution determined by the modelled plume injection heights, which are generally higher in FireWork-CFFEPS than FireWork-Ops. As noted in Section 3.1.4, FireWork-CFFEPS showed mixed forecast skill for hourly surface $PM_{2.5}$ for the Northern-Canada region, with over-predictions at the northernmost stations and under-predictions at the others, but in general FireWork-CFFEPS predicted higher surface $PM_{2.5}$ concentrations further away from fire locations. This is demonstrated in Figure 11 with the hourly surface $PM_{2.5}$ concentration comparisons for selected stations within the analysis area.

## 4. Discussion and Future Work

The model performance evaluation for the 2017 fire season presented in Section 3 gives us confidence in the improvements in model forecast skills that the new FireWork-CFFEPS system has over FireWork-Ops. Additional analysis for the most recent 2018 fire season, summarized in Section S3 of the Supplementary Material, showed similar, and consistent, changes when benchmarked against the operational FireWork-Ops system.

Despite the overall forecast improvements with FireWork-CFFEPS shown in Section 3 and the Supplementary Material, there are important science questions that remain to be investigated. Although we have quantified the changes in estimated fire emissions and in modelled plume injection heights, we have not independently verified these values. Verification of fire emission values is challenging as there are no direct measurements; nevertheless, it is possible to compare daily emissions predicted by FireWork-CFFEPS with many global fire emission inventories that implement the top-down, satellite derived FRP approach to estimate emission totals. Similarly, new techniques that calculate hotspot-specific fire $NO_2$, CO and $NH_3$ emissions directly from satellite measurements can provide comparable data for comparisons (Adams et al., 2019; Mebust et al., 2011). Ongoing studies will evaluate the modelled plume injection heights with available measurements such as those derived from the MISR instrument on the NASA TERRA satellite, as well as the higher resolution CALIPSO space instrument (Val Martin et al., 2018; Yao et al., 2018).

Lastly, a more quantitative comparison of satellite-derived aerosol optical depth (AOD) with modelled fire-$PM_{2.5}$ VCD can also help to examine the impact of modelled plume injection height and subsequent transport on forecast results.

Although FireWork-CFFEPS represents an important step forward in NRT modelling of fire emissions for regional air quality forecast systems, it still has some known limitations:

• Fire detection is from the operational CWFIS which is based on sensors on polar orbiting satellites. Although multiple sensors on multiple satellites are used, they still have limited temporal coverage of about six times a day. Fire starts after satellite overpasses at nadir (typically 1 pm LST) will not be considered until the next forecast simulation and detected fires are assumed to continue burning for the next forecast simulation day. Current hotspot retrievals are also limited by the presence of thick cloud or smoke, which can result in missing hotspot detection and hence missing fire emissions. Similarly, small fires with low heat

signatures, including prescribed burns or agricultural burning, may be undetected due to low sensor resolution. Prescribed burns make up a significant fraction of the U.S. $PM_{2.5}$ emissions inventory (Huang et al., 2018; Pouliot et al., 2017), but are negligible in Canada.

• Most fire behaviour models, including the FBP system used in CFFEPS, assume that fires grow freely without suppression.

• The same emission factors are now applied for all input fuel types in CFFEPS, but emission factors can vary by fuel type as found in recent measurements (Liu et al., 2017) and fuel-type-specific emission factors can be considered in future.

• Although fire growth is now closely tied to forecast hourly meteorology in CFFEPS, the key input, fire size or burn area per day, is still a pre-determined parameter that is based on an annual climatology of recorded fire area by province, and total number of hotspot retrievals. The daily fire size is also assumed to be persistent for the second-day forecast and is not based on

estimates from a fire-growth model driven by meteorology.

• Fire emissions are still treated as point sources and their location data are still assigned to model grid cells by geographic coordinates. This is necessary as fire injection height is specific to each hotspot. However, as fire area gets larger and model grid resolution becomes finer, grouping of fire hotspots as area aggregates may be a more favourable approach. This would allow spatial tracking of fire front by areas of flaming and smoldering combustion processes, and quantification of three-dimensional

fire growth over area and depth of burn.

One important limitation of the current model setup is the neglect of the interactions of fire behaviour with microphysics. Large, intense fires can affect local weather through release of surface heat flux and latent heat from water vapor. This energy can further increase the buoyancy of fire plumes, generating strong updrafts and accelerating the vertical transport of smoke. In the case of large fires, the increased buoyancy can even cause the formation of pyrocumulus or pyrocumulonimbus that further transport

smoke aloft, sometimes into the stratosphere. The differences in vertical wind shear resulting from higher injection heights can also alter the horizontal development of smoke plumes and impact the long-range transport of smoke.

Additionally, increases in primary PM emissions and in secondary aerosol formation in smoke plumes can increase atmosphere opacity, or AOD, along a plume trajectory. This can suppress turbulent mixing near the surface, causing stagnation and promoting the accumulation of surface pollutants. AOD increases can also attenuate overall photolysis rates, which may reduce

the chemical formation of surface $O_3$ and other atmospheric oxidants. The magnitudes of these meteorology and chemistry feedbacks are tightly coupled and are correlated to fire intensity. These complex feedback interactions can be important but are not yet considered in FireWork.

Active research is currently underway towards integrating CFFEPS directly into the research version of GEM-MACH model with direct and indirect two-way meteorology-chemistry feedbacks (Gong et al., 2015). Through close coupling of meteorology and

chemistry, and now with inputs of fire energy and emissions from CFFEPS enhancing vertical transport and impacting model microphysics, research using such an integrated system may provide the means to further examine the complex systems of direct and indirect feedbacks fire activities have on regional meteorology and chemistry.

## 5. Summary and Conclusions

FireWork is one of the first operational high-resolution regional air quality forecast systems with NRT wildfire emissions over a large North American domain. Since it became operational in 2016, the system has become an important guidance tool for air quality meteorologists in assessing potential air pollution episodes from the impact of forest fire smoke and issuing AQHI advisories for communities across Canada. In the initial development of the FireWork v1.0 system a number of compromises and assumptions were made to simplify the NRT wildfire emissions processing. In this work, we introduce a new, process-based, fire emission prediction system – CFFEPS – that has been integrated into FireWork to improve the representation of the dynamics of fire behaviour and smoke emissions while still ensuring the timely delivery of forecast products.

The new FireWork-CFFEPS (FireWork v2.0) system represents a significant step forward in the simulation of wildland fire smoke behaviour and fire emissions for regional CTMs. The changes listed in Section 2.4 have improved several aspects of fire emissions modelling and have resulted in better emission quantification. These changes include the introduction of location- and fuel-type-specific fire size, a revised North American fuel map, and updated emission factors. Fire emissions estimates are now process-based such that emission duration and temporal variation is driven by hourly meteorology and fuel depth-of-burn and its influence on combustion processes is considered. This approach allows for an improved application of combustion-phase-specific emission factors and more detailed chemical speciation that can be further extended by fuel-type dependence in the future. As well, fire-energy thermodynamics are now parameterized to calculate an hourly fire plume injection height that varies by fire size and fire intensity and that equilibrates with the hourly forecast environmental lapse rate at fire locations. The height of the modelled fire plume directly influences surface pollutant concentrations as well as long-range transport downwind of fire locations. The availability of hourly fire energy estimates also paves the way for on-going research on large wildfires as sources of heat energy for input to the microphysics scheme of the GEM-MACH coupled meteorology-chemistry model.

It is clear from the performance evaluation of the three AQ forecast systems reported here for three summer months in both 2017 and 2018 that the combined changes introduced in FireWork-CFFEPS have resulted in significant and consistent forecast improvements over FireWork-Ops for surface $PM_{2.5}$, and $O_3$ concentrations. Although the current FireWork-Ops is itself an improvement upon RAQDPS without NRT fire emissions, it tends to over-predict surface $PM_{2.5}$ and $O_3$ near hotspot locations. Forecast $PM_{2.5}$ surface concentrations are greatly improved with FireWork-CFFEPS despite higher estimated primary $PM_{2.5}$ emissions in some regions. The reason is that the new plume-injection height scheme has improved the vertical distribution of fire emission aloft, thus reducing over-predictions near fire hotspots, while at the same time reducing under-predictions for sites further away from fire hotspots with better treatments of long-range transport of fire smoke. Model performance statistics for regions with high fire activity showed overall improvements as well as better categorical scores for $PM_{2.5}$ and $O_3$ event-based concentrations. The statistics are slightly better for regions in Canada than those in the U.S. with lower errors and biases. For surface $NO_2$, on the other hand, there is less impact from fire activities due to lower emission contributions from fires compared to anthropogenic sources and shorter atmospheric lifetime, but unlike FireWork-Ops, the forecast results from FireWork-CFFEPS show no degradation to forecast skill from that of the RAQDPS system.

CFFEPS represents a new process-oriented approach to model fire emissions suitable for operational air quality forecasting as demonstrated with FireWork-CFFEPS. The process-based approach with bottom-up fire emissions estimates allows flexibility in

updating fuel-dependent emission factors and provides a more realistic yet computationally efficient plume-rise parameterization. Logistically, CFFEPS is also a bridge that brings together the science developed by two Canada federal departments, such that ECCC is able to access and to utilize the state-of-science fire behaviour research of the CFS and to couple the CFFEPS system with the latest understanding in meteorology and atmospheric chemistry embodied in ECCC's two operational air quality forecast

5    systems, the RAQDPS and FireWork.

**Code and data availability**

The air quality monitoring data used for model evaluation is available for download from the Canadian National Air Pollution Surveillance (NAPS) Network, and the U.S. Air Quality System (AQS) data repositories through the Internet URLs provided in Section 3. The code for CFFEPS v2.03 and the accompanying user manual are available from the Zenodo website:
https://doi.org/10.5281/zenodo.2579383. The GEM-MACH atmospheric chemistry module for the GEM (meteorology) numerical weather prediction model (Copyright ©2007–2013, Air Quality Research Division and National Prediction Operations Division, Environment and Climate Change Canada) can be downloaded from the Zenodo website: https://doi.org/10.5281/zenodo.2579386. The CFFEPS and GEM-MACH codes are released as free software that can be redistributed and/or modified under the terms of the GNU Lesser General Public License, either version 2.1 or any later version,
as published by the Free Software Foundation. The GEM (meteorology) code is available for download from website https://github.com/mfvalin/gem. The executable for GEM-MACH is obtained by providing the chemistry library to GEM when generating its executable. All other data (model simulation outputs, emissions inputs) are available upon request from the corresponding author Jack Chen (jack.chen@canada.ca).

**Author contribution**

JC led the development of the ECCC FireWork system. KA is the principle developer of CFFEPS. JC, KA and RP designed the FireWork-CFFEPS model framework. MDM is responsible for development of the RAQDPS operational system and for reviewing this study. RMA contributed the emission factors used in the current FireWork-CFFEPS study. PE provided CWFIS integrations for CFFEPS, calculation of hotspot areas, and all hotspot data files used in hindcast simulations. DT provided the CFFEPS fuel map integration for the North America model domain. RMA and HL were responsible for code integration and
model simulations. JC conducted the majority of analyses and model evaluation for this study. JC and KA prepared the manuscript with contributions from all co-authors.

**Supplement**

The supplement related to this article is available:

**Competing interests**

The authors declare that they have no conflict of interest.

**Acknowledgement**

The authors gratefully acknowledge the near-real-time fire information retrievals from the NOAA National Environmental Satellite, Data and Information Service (NESDIS), the NASA Fire Information for Resource Management System (FIRMS), the USFS GTAC and University of Maryland VIIRS active fire group. We also thank Samuel Gilbert (ECCC) for his assistance in
using the Verification of Air Quality Models (VAQUM) database system for data analysis, Sétigui Keita (ECCC) for generating the monthly mean fire-$PM_{2.5}$ concentration contours, Junhua Zhang (ECCC) for producing the anthropogenic emission summary tables and Paul Makar (ECCC) for organizing the Forest Fire Working Group to promote and facilitate wildfire research and collaboration across Canadian government agencies. Lastly, we acknowledge the helpful comments from the two anonymous reviewers.

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

**Tables and Figures**

**Table 1. FBP fuel types and bulk densities (g cm⁻³) of surface fuels for each fuel type.**

| Fuel type | 0-2 cm | 2-4 cm | 4-6 cm | 6-8 cm[1] | Fuel type description |
|---|---|---|---|---|---|
| C-1 | 0.045 | | | | Spruce-Lichen Woodland[2] |
| C-2 | 0.019 | 0.034 | 0.051 | 0.056 | Boreal Spruce |
| C-3 | 0.015 | 0.020 | 0.032 | 0.066 | Mature Jack or Lodgepole Pine |
| C-4 | 0.022 | 0.029 | 0.045 | 0.059 | Immature Jack or Lodgepole Pine |
| C-5 | 0.093 | | | | Red and White Pine |
| C-6 | 0.030 | 0.050 | 0.050 | 0.050 | Conifer Plantation |
| C-7 | 0.100 | 0.100 | 0.050 | 0 | Ponderosa Pine-Douglas-Fir |
| D-1 | 0.061 | | | | Leafless Aspen |
| M-1/2 | 0.0265 | 0.071 | 0.0795 | 0.082 | Boreal Mixwood (spring/summer) |
| M-3/4 | 0.041 | 0.061 | 0.084 | 0.112 | Dead Balsam Fir Mixedwood (spring/summer) |
| O-1 | GFL | | | | Grass[3] |
| S-1 | 0.200 | | | | Jack or Lodgepole Pine Slash[4] |
| S-2 | 0.500 | | 0.300 | | White spruce-balsam Slash |
| S-3 | 0.600 | | 1.000 | | Coastal Cedar-Hemlock-Douglas Fir Slash |

[1] bulk density below 8 cm assumed to equal that of 6-8 cm layer
[2] Fuel depth for C-1 is typically less than 2 cm
[3] Grass fuel load (typically 0.3 kg m⁻²)
[4] Slash consists of cut tree tops and branches after clearcut logging

**Table 2. Combustion stage allocation factors for each fuel type.**

| Fuel | | Flaming | Smoldering | Residual |
|---|---|---|---|---|
| Ground | Litter | 0.90 | 0.10 | 0 |
| | Upper duff | 0.10 | 0.70 | 0.20 |
| | Lower duff | 0 | 0.20 | 0.80 |
| Canopy | | 0.94 | 0.06 | 0 |
| Slash | | 0.70 | 0.15 | 0.15 |
| Grass | | 0.95 | 0.05 | 0 |

10  **Table 3. Emission factors (g kg⁻¹) of species currently used in CFFEPS.**

| Species | Flaming | Smoldering | Residual |
|---|---|---|---|
| $PM_{10}$ | 16.05 | 27.38 | 40.75 |
| $PM_{2.5}$ | 13.6 | 23.2 | 34.53 |
| CO | 83 | 135 | 248 |
| $CH_4$ | 3.23 | 7.32 | 9.94 |
| NMHC | 19.85 | 33.87 | 56.08 |
| $NO_x$ | 1.83 | 2 | 0.45 |
| $NH_3$ | 0.99 | 1.5 | 1.94 |
| $SO_2$ | 0.93 | 1.06 | 1.76 |

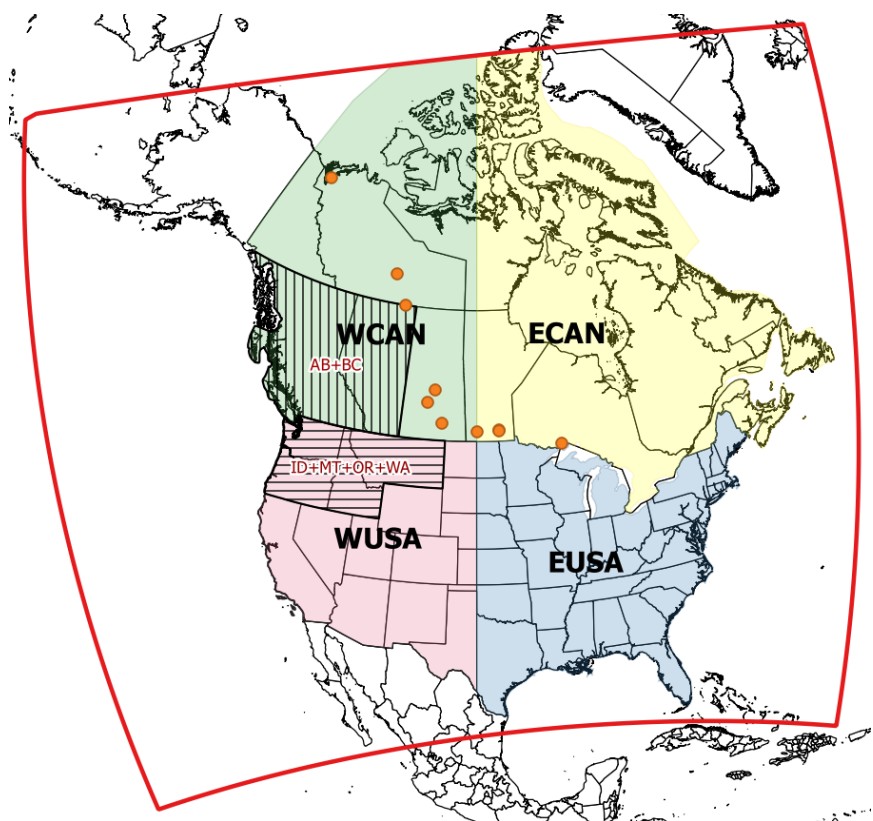

**Figure 1. The boundary of the FireWork and RAQDPS domain is indicated by the red outline. The four continental subregions (ECAN, EUSA, WCAN, WUSA) considered for model performance evaluation are denoted by different colours. The spatial extent of AB+BC and ID+MT+OR+WA, the two wildfire source regions used for model performance evaluation, are indicated by different diagonal hatching. The locations of the 10 AQ measurement stations in the Northern-Canada region are marked by orange dots (see also Table S3).**

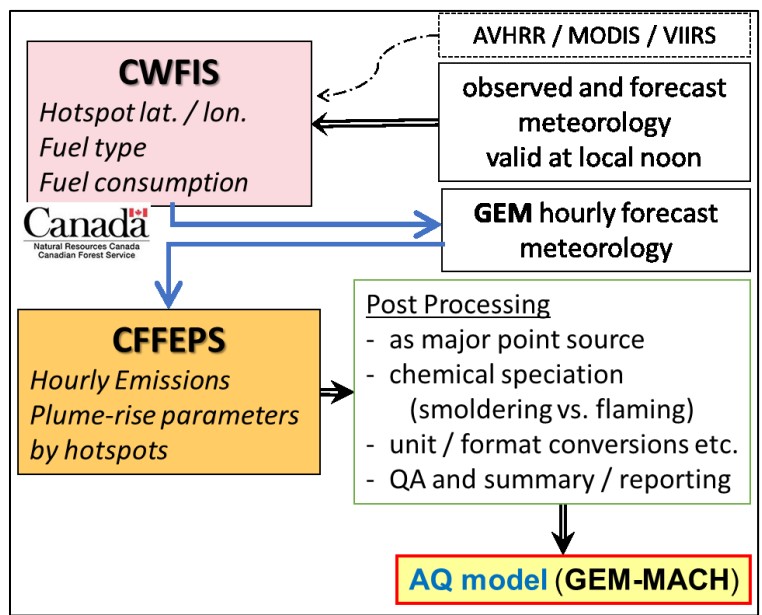

**Figure 2. Structure and data flow for the integration of CFFEPS with GEM-MACH.**

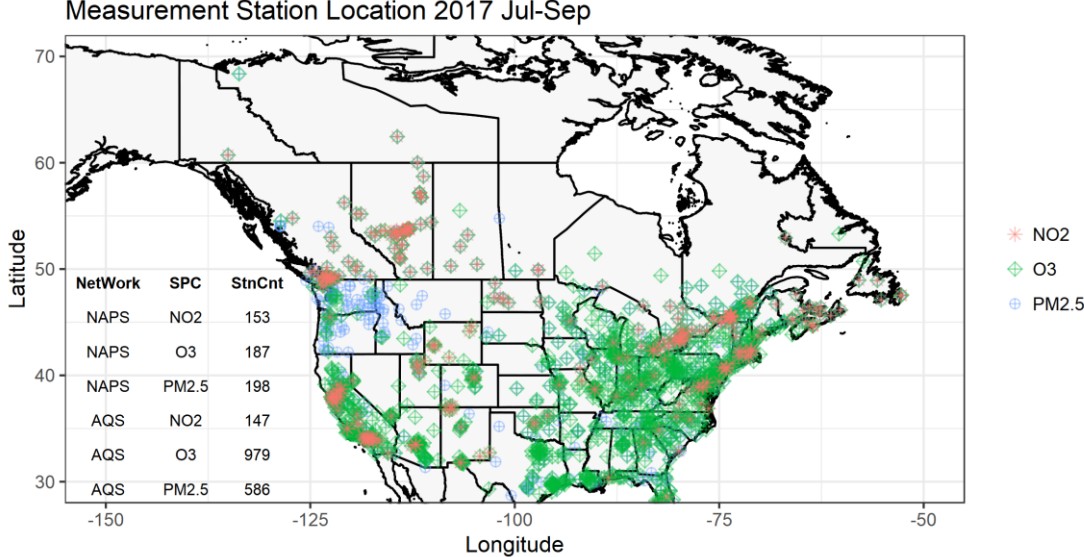

**Figure 3. Locations of Canadian NAPS and U.S. AQS stations used for 2017 model evaluation with 75% measurement completeness criteria. Some stations measure more than one species.**

**Table 4. Categorical score definitions for a binary event.**

| Observed/Forecasted | NO | YES | |
|---|---|---|---|
| NO | Correct Non-event | False Alarm | Not Observed |
| YES | Miss | Hit | Observed |
| | Not Forecasted | Forecasted | TOTAL |

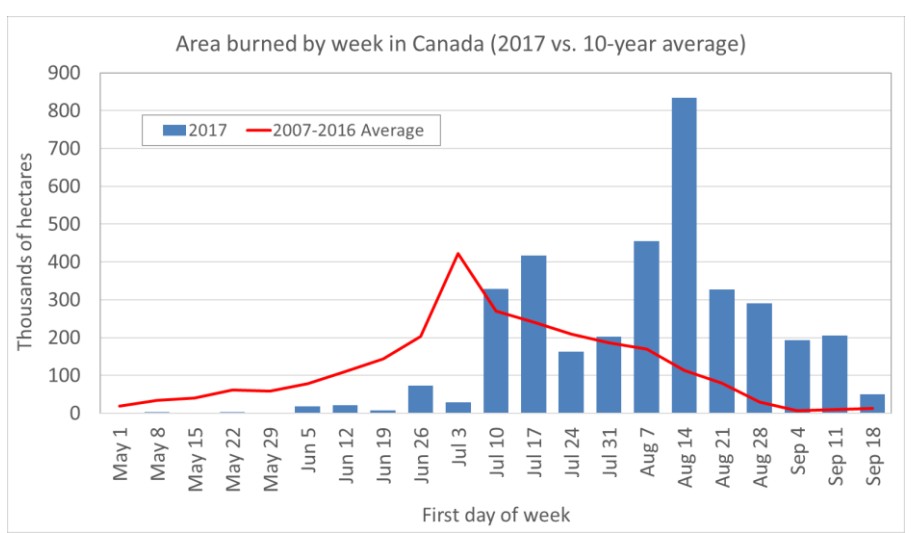

**Figure 4. National fire burn area in Canada by week starting for 2017 fire season (blue vertical bars) and previous 10-year average (red line).**

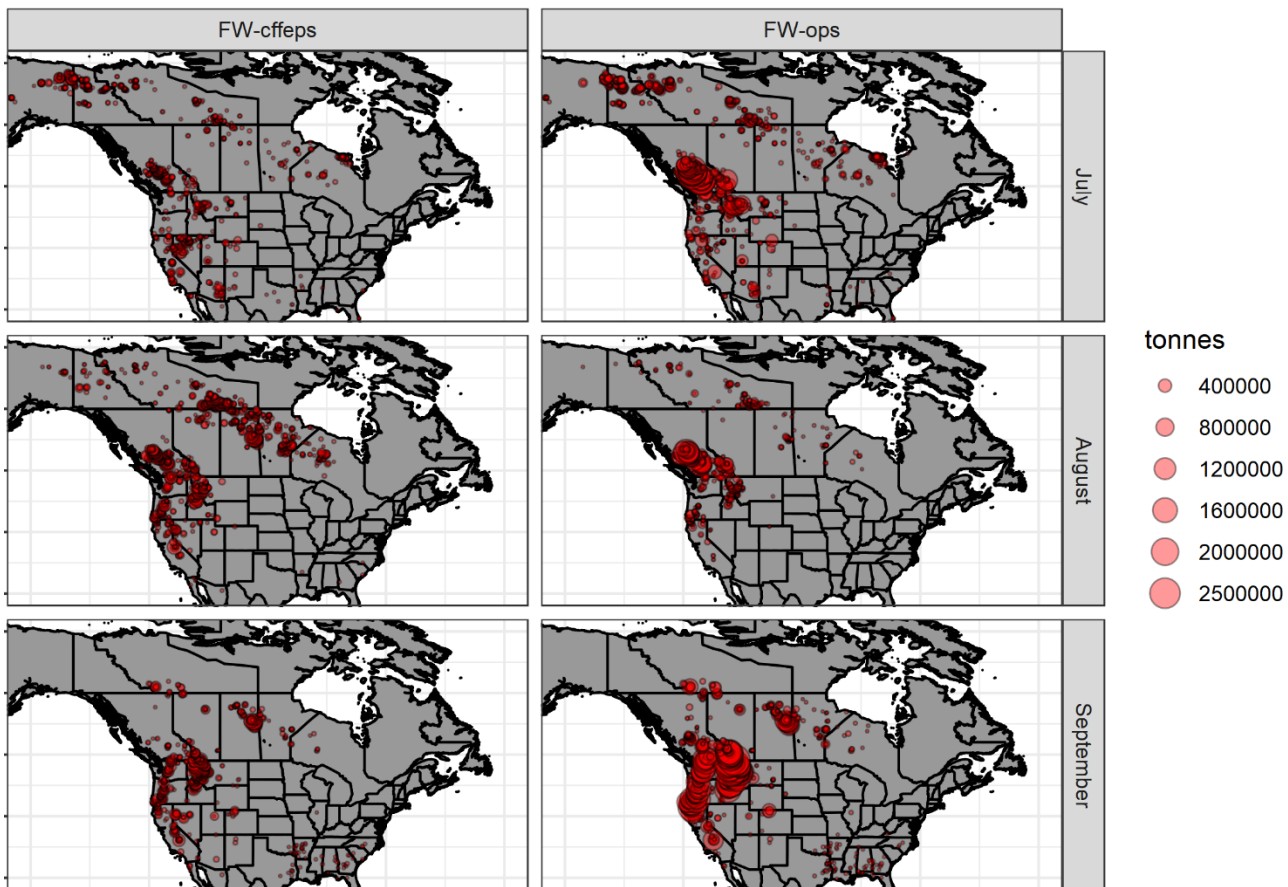

**Figure 5. Monthly total effective biomass burned by hotspot for FireWork-CFFEPS (left) and FireWork-Ops (right) for fires with threshold greater than 5,000 tonnes/month.**

**Table 5. Total fire emissions (kilotonne) in Canada, the U.S., and selected provinces and states with high fire activity from FireWork-Ops for the July-Sept. 2017 period and percentage differences (in italics) for FireWork-CFFEPS.**

| | VOC | NOX | CO | PM2.5 | PM10 | NH3 | SO2 |
|---|---|---|---|---|---|---|---|
| CAN | 1,208 | 159 | 11,572 | 1,043 | 1,231 | 191 | 110 |
| USA | 802 | 125 | 7,653 | 711 | 839 | 127 | 81 |
| *CAN* | *-34%* | *-40%* | *7%* | *88%* | *88%* | *-35%* | *-6%* |
| *USA* | *-33%* | *-47%* | *9%* | *87%* | *87%* | *-34%* | *-14%* |

| | VOC | NOX | CO | PM2.5 | PM10 | NH3 | SO2 |
|---|---|---|---|---|---|---|---|
| CA.AB | 12.4 | 1.5 | 120 | 10.6 | 12.5 | 2.0 | 1.1 |
| CA.BC | 767.2 | 109.2 | 7338 | 670.3 | 790.9 | 121.6 | 73.1 |
| CA.MB | 45.1 | 5.3 | 433 | 38.3 | 45.2 | 7.1 | 3.8 |
| CA.NT | 226.0 | 22.0 | 2180 | 187.7 | 221.5 | 35.8 | 17.1 |
| CA.ON | 23.5 | 3.0 | 225 | 20.2 | 23.9 | 3.7 | 2.1 |
| CA.SK | 117.0 | 16.6 | 1119 | 102.2 | 120.6 | 18.5 | 11.1 |
| | | | | | | | |
| US.WA | 111.2 | 18.4 | 1059 | 99.6 | 117.5 | 17.6 | 11.7 |
| US.OR | 132.2 | 16.4 | 1269 | 113.1 | 133.5 | 20.9 | 11.5 |
| US.MT | 255.3 | 48.0 | 2421 | 234.2 | 276.4 | 40.5 | 29.5 |
| US.ID | 135.5 | 24.0 | 1288 | 122.9 | 145.1 | 21.5 | 15.0 |

| | VOC | NOX | CO | PM2.5 | PM10 | NH3 | SO2 |
|---|---|---|---|---|---|---|---|
| *CA.AB* | *73%* | *73%* | *183%* | *404%* | *404%* | *72%* | *163%* |
| *CA.BC* | *-63%* | *-75%* | *-36%* | *7%* | *7%* | *-64%* | *-49%* |
| *CA.MB* | *13%* | *21%* | *90%* | *242%* | *242%* | *16%* | *77%* |
| *CA.NT* | *-10%* | *18%* | *40%* | *159%* | *159%* | *-13%* | *52%* |
| *CA.ON* | *-4%* | *-1%* | *58%* | *184%* | *184%* | *-2%* | *40%* |
| *CA.SK* | *51%* | *25%* | *156%* | *342%* | *342%* | *53%* | *109%* |
| | | | | | | | |
| *US.WA* | *-60%* | *-76%* | *-29%* | *16%* | *16%* | *-60%* | *-50%* |
| *US.OR* | *-48%* | *-60%* | *-10%* | *53%* | *53%* | *-49%* | *-21%* |
| *US.MT* | *-47%* | *-70%* | *-9%* | *47%* | *47%* | *-47%* | *-40%* |
| *US.ID* | *-48%* | *-67%* | *-10%* | *47%* | *47%* | *-48%* | *-38%* |

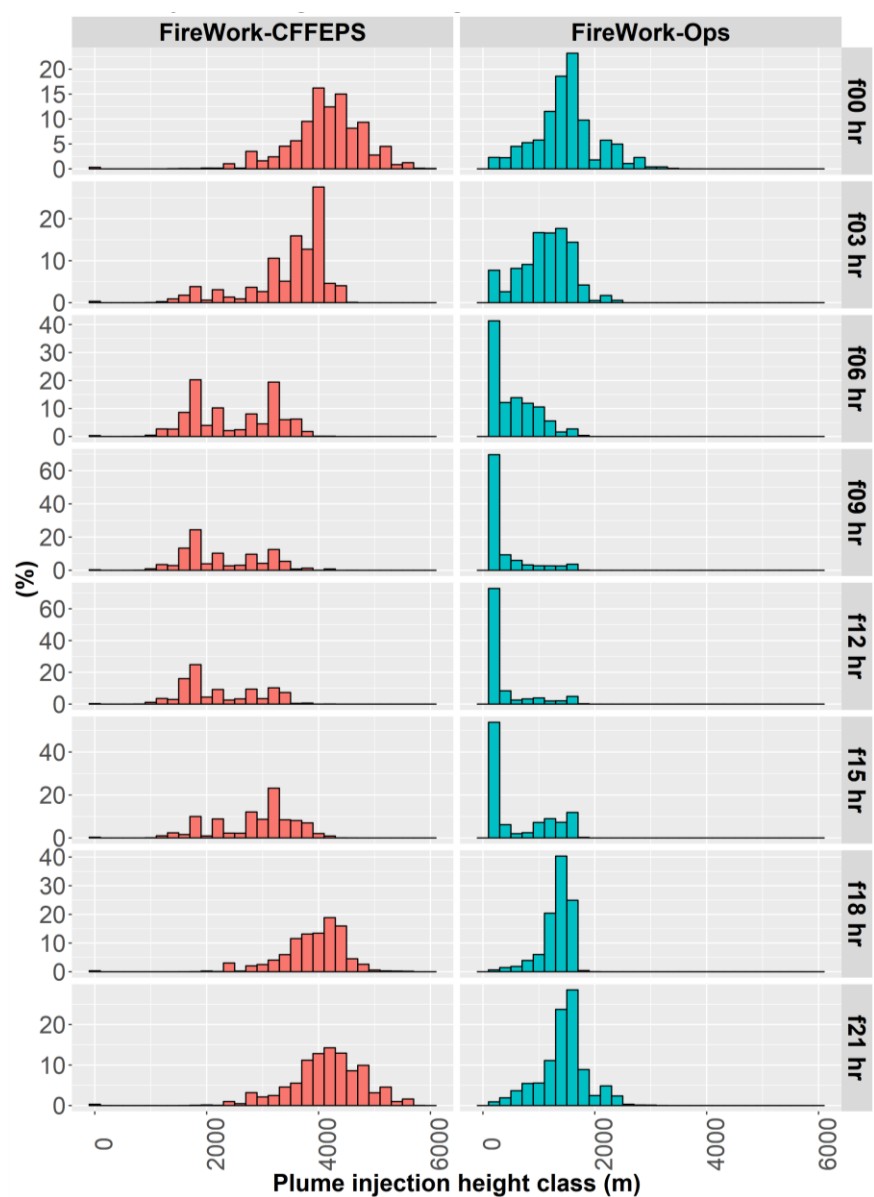

**Figure 6. Modelled plume injection height frequency distribution by 200 m altitude intervals for all fires in British Columbia, Canada in August 2017 by forecast hour for 00 UTC forecasts by FireWork-CFFEPS (left) and FireWork-Ops (right).**

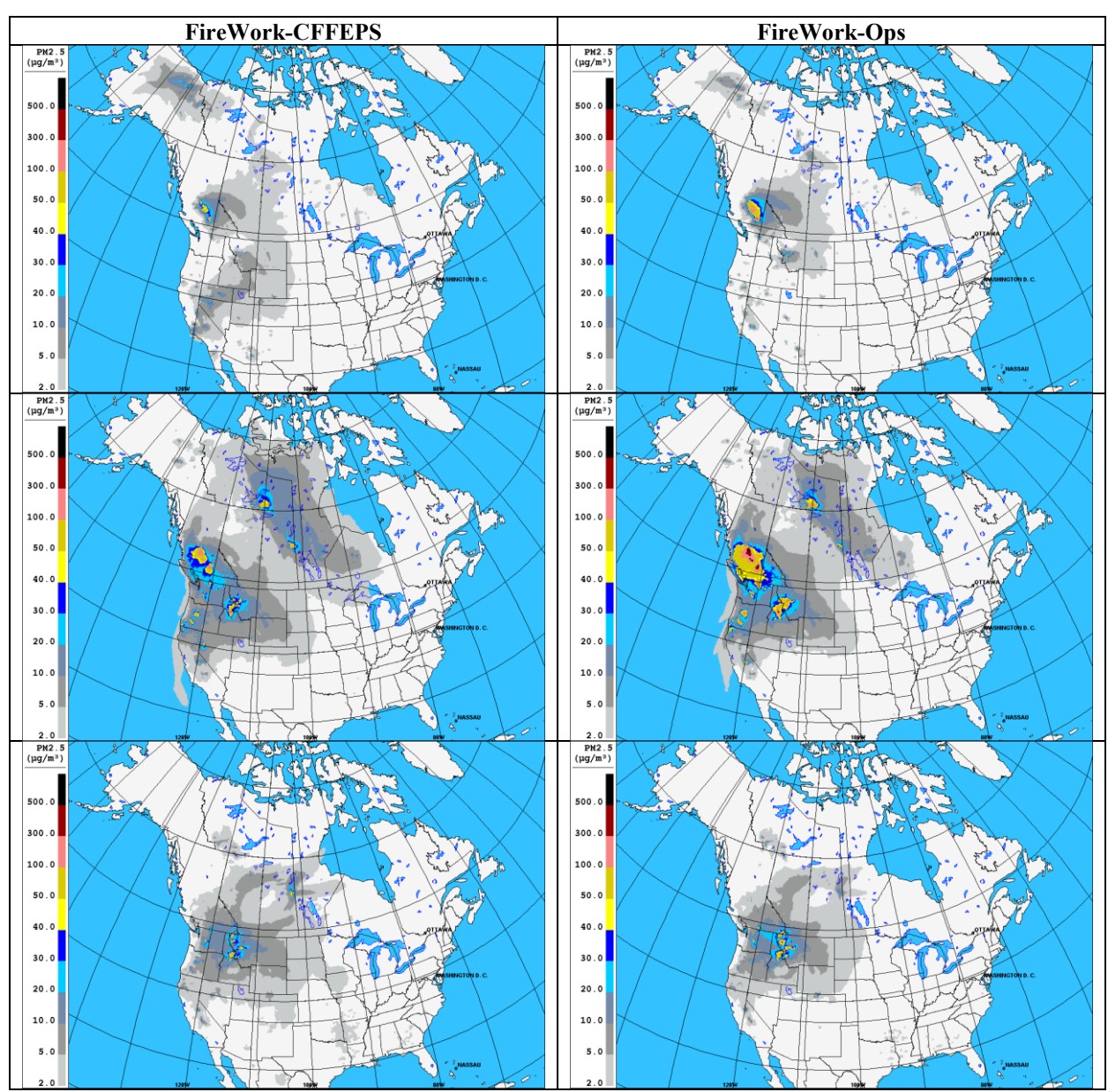

**Figure 7. Mean monthly surface fire-PM$_{2.5}$ concentrations (μg m$^{-3}$) from FireWork-CFFEPS (left) and FireWork-Ops (right) for July (top row), August (middle row), and September (bottom row) 2017.**

**Table 6. Model performance statistics for daily maximum surface (a) PM$_{2.5}$, (b) O$_3$, and (c) NO$_2$ concentrations for July-Sept. 2017 for three modelling systems for four geographic regions (shown in Figure 1). Units of Ō, M̄, MB, and RMSE are µg m$^{-3}$ for PM$_{2.5}$ and ppbv for O$_3$ and NO$_2$.**

| (a) PM$_{2.5}$ | WCAN (88 stations, n=7849) | | | ECAN (110 stations, n=9478) | | | WUSA (221 stations, n=19504) | | | EUSA (365 stations, n=31570) | | |
|---|---|---|---|---|---|---|---|---|---|---|---|---|
| Model | RAQDPS | FW-Ops | FW-CFFEPS | RAQDPS | FW-Ops | FW-CFFEPS | RAQDPS | FW-Ops | FW-CFFEPS | RAQDPS | FW-Ops | FW-CFFEPS |
| Ō | 23 | | | 14 | | | 23 | | | 17 | | |
| M̄ | 12 | 44 | 29 | 15 | 16 | 17 | 15 | 44 | 31 | 18 | 19 | 19 |
| MB | -11.1 | 21.1 | 5.8 | 1.7 | 2.3 | 2.9 | -8.0 | 21.1 | 8.6 | 1.9 | 2.1 | 2.8 |
| R | 0.15 | 0.57 | 0.64 | 0.24 | 0.26 | 0.26 | 0.17 | 0.49 | 0.59 | 0.23 | 0.23 | 0.24 |
| RMSE | 32 | 101 | 39 | 16 | 16 | 16 | 29 | 164 | 64 | 14 | 14 | 14 |

| (b) O$_3$ | WCAN (73 stations, n=6679) | | | ECAN (114 stations, n=10397) | | | WUSA (289 stations, n=26317) | | | EUSA (690 stations, n=62187) | | |
|---|---|---|---|---|---|---|---|---|---|---|---|---|
| Model | RAQDPS | FW-Ops | FW-CFFEPS | RAQDPS | FW-Ops | FW-CFFEPS | RAQDPS | FW-Ops | FW-CFFEPS | RAQDPS | FW-Ops | FW-CFFEPS |
| Ō | 39 | | | 37 | | | 59 | | | 46 | | |
| M̄ | 44 | 59 | 44 | 46 | 47 | 45 | 71 | 76 | 70 | 68 | 68 | 66 |
| MB | 4.1 | 19.1 | 4.7 | 9.5 | 10.2 | 8.2 | 12.1 | 16.5 | 11.1 | 21.5 | 22.0 | 20.4 |
| R | 0.62 | 0.38 | 0.58 | 0.78 | 0.78 | 0.75 | 0.67 | 0.54 | 0.64 | 0.66 | 0.66 | 0.65 |
| RMSE | 14 | 52 | 17 | 14 | 15 | 14 | 22 | 33 | 23 | 27 | 27 | 26 |

| (c) NO$_2$ | WCAN (78 stations, n=7156) | | | ECAN (75 stations, n=6869) | | | WUSA (78 stations, n=7073) | | | EUSA (78 stations, n=6250) | | |
|---|---|---|---|---|---|---|---|---|---|---|---|---|
| Model | RAQDPS | FW-Ops | FW-CFFEPS | RAQDPS | FW-Ops | FW-CFFEPS | RAQDPS | FW-Ops | FW-CFFEPS | RAQDPS | FW-Ops | FW-CFFEPS |
| Ō | 12 | | | 11 | | | 15 | | | 13 | | |
| M̄ | 18 | 19 | 18 | 18 | 18 | 18 | 30 | 30 | 29 | 28 | 28 | 27 |
| MB | 5.5 | 6.7 | 5.1 | 7.6 | 7.7 | 7.3 | 14.1 | 14.4 | 13.3 | 14.7 | 14.8 | 14.2 |
| R | 0.59 | 0.55 | 0.59 | 0.73 | 0.73 | 0.73 | 0.67 | 0.67 | 0.68 | 0.67 | 0.67 | 0.66 |
| RMSE | 13 | 16 | 13 | 14 | 14 | 14 | 22 | 22 | 21 | 22 | 22 | 21 |

**Table 7. Categorical scores for July-September 2017 by geographic region for hourly (a) PM$_{2.5}$ events or exceedances based on a threshold of 30 μg m$^{-3}$, (b) O$_3$ events based on a threshold of 65 ppbv, and (c) NO$_2$ events based on a threshold of 30 ppbv.**

| (a) PM$_{2.5}$ | WCAN | | | WUSA | | |
|---|---|---|---|---|---|---|
| Model | RAQDPS | FW-Ops | FW-CFFEPS | RAQDPS | FW-Ops | FW-CFFEPS |
| POD | 7% | 65% | 59% | 5% | 64% | 59% |
| FAR | 61% | 48% | 43% | 82% | 48% | 47% |
| CSI | 6% | 40% | 41% | 4% | 40% | 39% |

| (b) O$_3$ | WCAN | | | WUSA | | |
|---|---|---|---|---|---|---|
| Model | RAQDPS | FW-Ops | FW-CFFEPS | RAQDPS | FW-Ops | FW-CFFEPS |
| POD | 48% | 74% | 58% | 60% | 65% | 58% |
| FAR | 86% | 95% | 88% | 58% | 62% | 58% |
| CSI | 12% | 5% | 11% | 33% | 31% | 32% |

| (c) NO$_2$ | WCAN | | | WUSA | | |
|---|---|---|---|---|---|---|
| Model | RAQDPS | FW-Ops | FW-CFFEPS | RAQDPS | FW-Ops | FW-CFFEPS |
| POD | 49% | 49% | 49% | 76% | 76% | 75% |
| FAR | 93% | 93% | 92% | 88% | 88% | 88% |
| CSI | 7% | 6% | 7% | 11% | 11% | 12% |

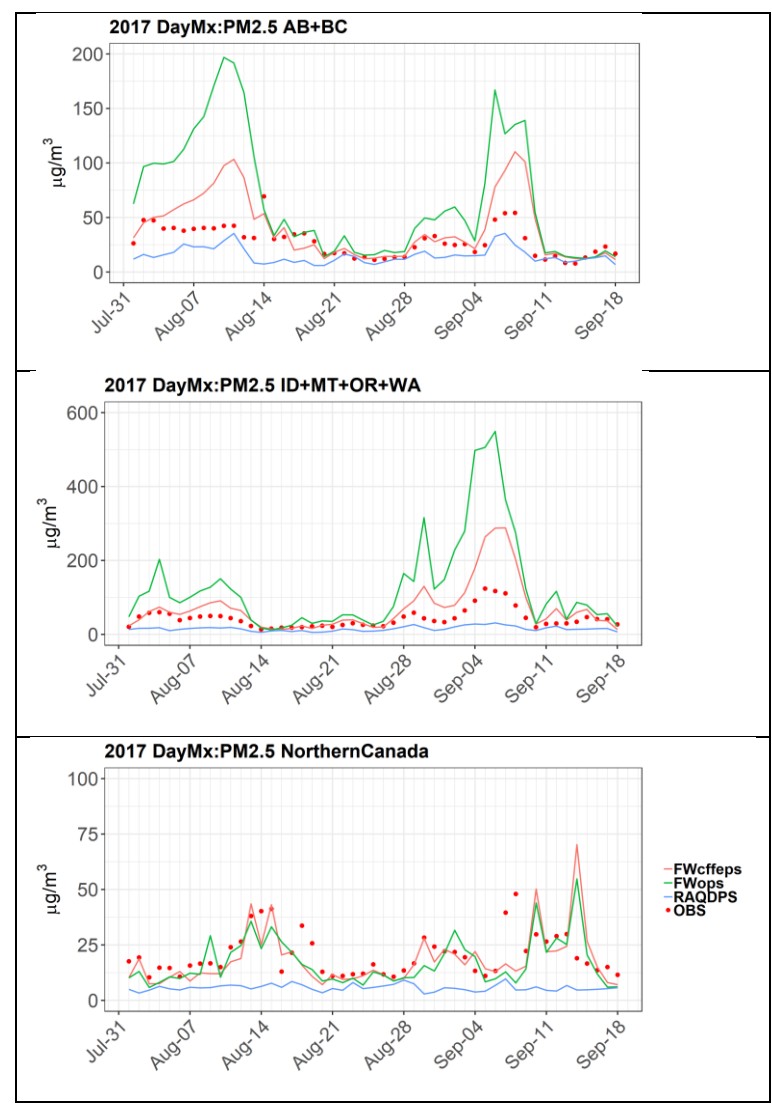

**Figure 8. Time series of mean daily maximum PM$_{2.5}$ concentration (μg m$^{-3}$) from Aug. 1 to Sept. 18, 2017 for the three forecast models and surface measurements (OBS) averaged across measurement stations in the AB+BC, ID+MT+OR+WA, and Northern-Canada regions.**

**Table 8. Model performance statistics for daily maximum surface PM$_{2.5}$ concentration (µg m$^{-3}$) for Aug. 1 to Sept. 18, 2017 for stations in AB+BC, ID+MT+OR+WA, and Northern-Canada regions.**

| PM$_{2.5}$ | AB+BC (79 stations) | | | ID+MT+OR+WA (89 stations) | | | Northern-Canada (10 stations) | | |
|---|---|---|---|---|---|---|---|---|---|
| Model | RAQDPS | FW-Ops | FW-CFFEPS | RAQDPS | FW-Ops | FW-CFFEPS | RAQDPS | FW-Ops | FW-CFFEPS |
| count | 3776 | | | 4257 | | | 440 | | |
| Ō | 28 | | | 42 | | | 20 | | |
| M̄ | 15 | 66 | 40 | 15 | 126 | 71 | 6 | 17 | 19 |
| MB | -13 | 38 | 12 | -27 | 84 | 29 | -15 | -3 | -1 |
| R | 0.18 | 0.59 | 0.63 | 0.19 | 0.47 | 0.60 | 0.09 | 0.54 | 0.52 |
| RMSE | 36 | 133 | 44 | 52 | 342 | 121 | 30 | 24 | 26 |

**Table 9. PM$_{2.5}$ categorical scores based on a threshold of 30 µg m$^{-3}$ for Aug. 1 to Sept. 18, 2017 for stations in the AB+BC, ID+MT+OR+WA, and Northern-Canada regions.**

| PM$_{2.5}$ | AB+BC | | | ID+MT+OR+WA | | | Northern-Canada | | |
|---|---|---|---|---|---|---|---|---|---|
| Model | RAQDPS | FW-Ops | FW-CFFEPS | RAQDPS | FW-Ops | FW-CFFEPS | RAQDPS | FW-Ops | FW-CFFEPS |
| POD | 9% | 74% | 68% | 3% | 72% | 66% | 0% | 24% | 22% |
| FAR | 53% | 46% | 42% | 59% | 37% | 34% | Inf. | 58% | 58% |
| CSI | 8% | 45% | 46% | 3% | 50% | 50% | 0% | 17% | 17% |

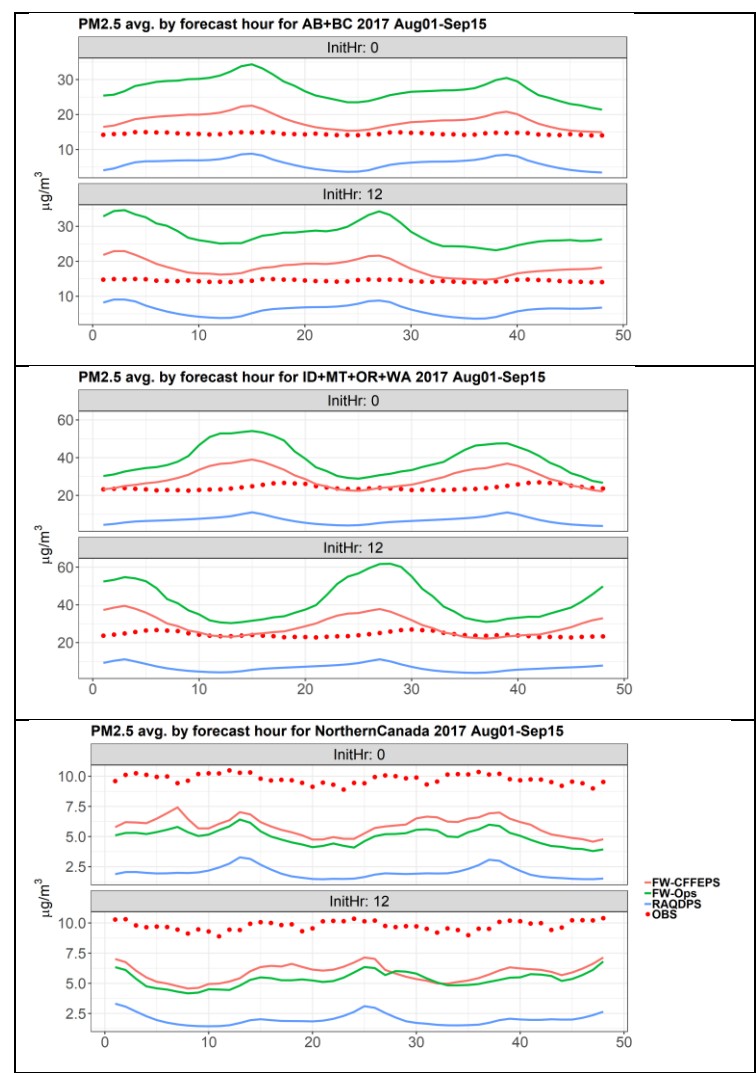

**Figure 9. Mean PM₂.₅ concentrations (μg m⁻³) by forecast hour (f00–f48) for the period from Aug. 1 to Sept. 16, 2017 for the three forecast models and surface measurements (OBS) for AQ measurement stations in the AB+BC (top), ID+MT+OR+WA (centre), and Northern-Canada (bottom) regions.  The forecasts launched at 00 UTC and 12 UTC are analysed separately.**

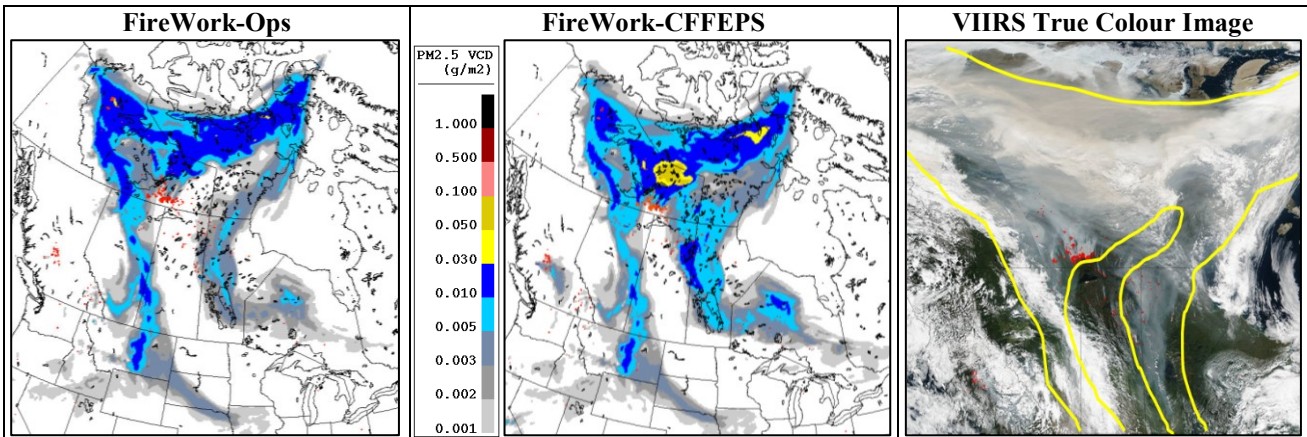

**Figure 10. Fire-PM$_{2.5}$ VCD (g m$^{-2}$) forecast by FireWork-Ops (left) and FireWork-CFFEPS (middle) from the Aug. 13, 2017 00 UTC forecast run valid for Aug. 14, 2017 12UTC, and (right) VIIRS true colour satellite image for Aug. 14 2017 with lines superimposed to aid comparison. Fire hotspots are represented in red. VIIRS image source: NASA**
5    **https://www.nasa.gov/image-feature/goddard/2017/smoke-and-clouds-obscure-skies-in-northern-canada**

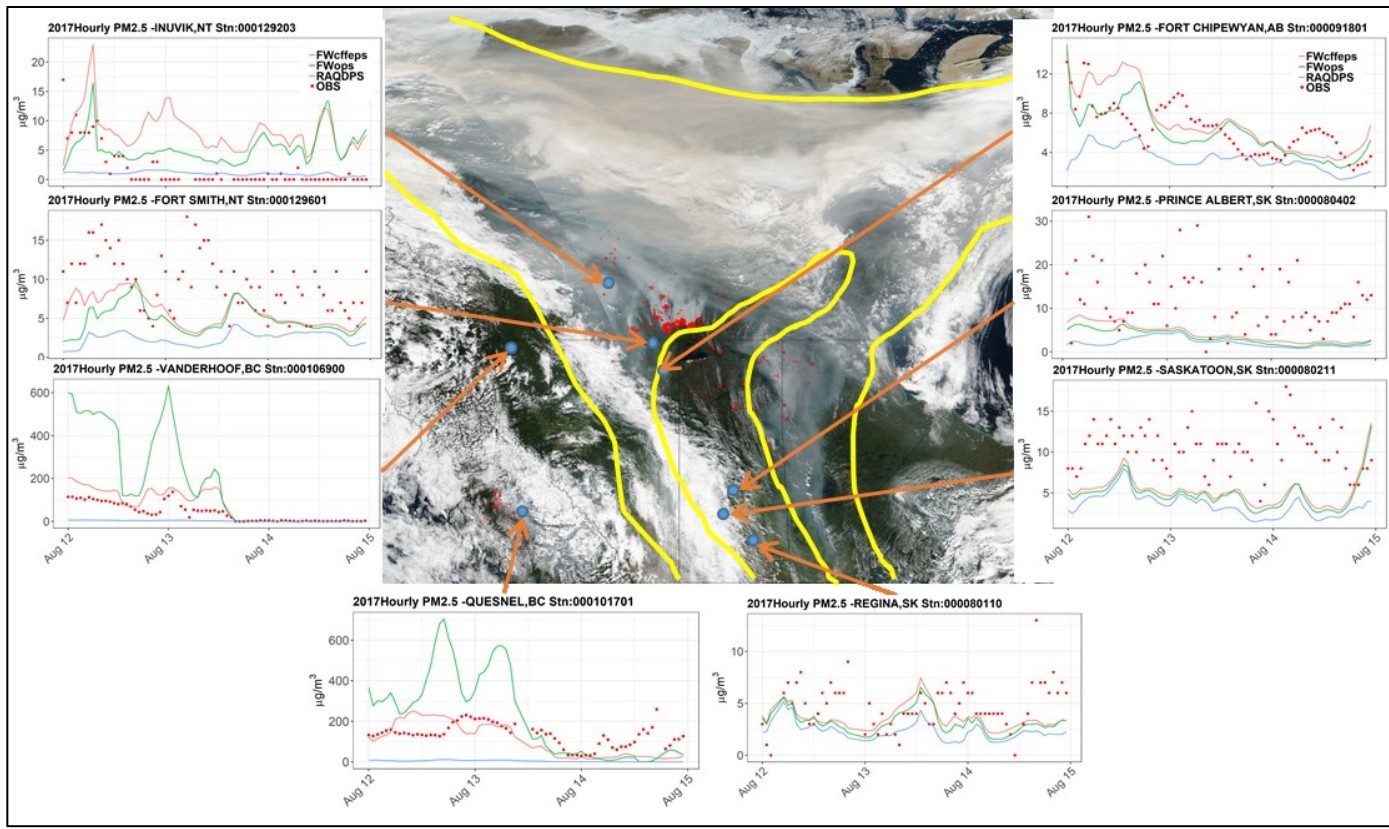

**Figure 11. Time series of hourly surface PM₂.₅ concentations (μg m⁻³) from Aug. 12–15, 2017 for three model simulations and measurements (OBS) for selected stations with station locations identified by red arrows, and (centre) VIIRS true colour satellite image for Aug. 13, 2017 with yellow lines superimposed to aid comparison. Fire hotspots are represented in red dots in VIIRS image. VIIRS image source: NASA https://www.nasa.gov/image-feature/goddard/2017/smoke-and-clouds-obscure-skies-in-northern-canada**