# Peer review of "The FireWork v2.0 air quality forecast system with biomass burning emissions from the Canadian Forest Fire Emissions Prediction System v2.03"

_Geoscientific Model Development, 2019_

## Referee Comment (RC1) · Anonymous Referee #1 · 16 Apr 2019

The authors compare modeling system predictions of PM2.5, O3, and NO2 against routine surface measurement sites from 3 different forecasting systems: 1) a system with no wildfire, 2) the existing version of the Canadian forecasting system, and 3) the newly updated Canadian forecasting system. Simulations 2 and 3 include wildfire emissions, which are treated differently in each system. Multiple enhancements to simulating wildfire emissions were implemented in the new forecasting system compared to the existing system including emission factors, plume height, and vertical distribution of smoke emissions within the plume.

[Figure]

Wildfire impacts are important to provide so that the public can be better informed about potential or actual exposure to change behavior to try and minimize exposure. The work presented here is very relevant and important for the broader modeling community and providing confidence in the approach is important toward establishing confidence in the forecasting product being provided to the public to understand reliability in the forecast. Overall, the changes made to the approach for estimating wildfire emissions make sense and would seem to favor an improved forecast. It is not totally clear which changes make the most impact since they were not systematically evaluated. However, the main concern is that it is not clear based on the information provided how well the system is performing for capturing smoke impacts. This is important work and very useful information that should be published. Comments that follow are intended to provide a stronger evaluation to more clearly isolate how the improvements in the forecasting system improve performance, which is a challenge since many monitors will typically be largely impacted by non-wildfire sources.

Too many monitor locations and too long of a period were aggregated, which really limits our understanding of how much better the new system is at predicting smoke impacts on PM2.5, O3, and NO2. A clearer demonstration (maybe in addition to the broad evaluation provided) would be to focus on a specific area of monitors that were known to be impacted for a few days or weeks by wildfire and see how well each of the modeling systems capture this known episode. Another approach would be to provide the monitor comparisons only when the model is predicting smoke impacts (whether either of the systems with wildfire emissions predicted smoke impacts). The way the information is presented it is not clear at all how well each system captures fire impacts, in particular for O3 and NO2. This is because the model performance is likely dominated by other sources or regions and not by the wildfire and the new approaches for estimating smoke may simply be compensating for underrepresentation of other sources. So if the authors can focus on monitors with periods dominated by wildfire impacts that would help make the evaluation much clearer.

[Figure]

The literature review in the introduction misses a few newer papers that looked at the performance of the Briggs approach. In these papers the Briggs approach performed reasonably for wildfire when provided realistic information about the fire size and timing. It does not appear that the approach limits plume tops to the modeled PBL height or uniformly distributes emissions through the plume. These may simply be choices made by the model developers for their implementation of Briggs but it begs the question about why these limitations were imposed in this system (or the older version of the system). Briggs is a component of the CMAQ model used for air quality forecasts by NOAA so references for that system may be relevant for the modified Briggs approach (Baker et al., 2018; Zhou et al., 2018) and a very brief note that the modified Briggs has been implemented differently in other systems.

Using 2010 Canadian emissions and 2011 U.S. emissions to represent 2017 (and later) will result in an overestimation of pollutants due to significant reductions in mobile source emissions (fleet turnover, emissions standard in the U.S. implemented in 2017, and fuel standard implemented in the U.S. in 2017) during this period, plant shutdowns, and fuel switching by EGUs from coal to natural gas. The emissions should be updated for the forecasting system and adjusted to reflect broader sector reductions. For the purposes of this manuscript and evaluation, the authors do not need to re-do the entire evaluation with newer emissions but need to clearly recognize that the emissions are likely leading to large overestimates of some species (like NOX in particular) which confounds the evaluation since wildfire NOX may not make performance "look better".

It is not clear which emissions are used for each system. Are fuel and combustion specific emissions used in any of the simulations? Text suggests they may be in the analysis but other text and the Tables suggest otherwise. What is the source of the emission factors? Please provide the speciation profiles for the different combustion components for VOC and PM2.5 in the supporting information with a citation as that is important and would be valuable to others modeling wildfire.

References
Baker, K., Woody, M., Valin, L., Szykman, J., Yates, E., Iraci, L., Choi, H., Soja, A., Koplitz, S., Zhou, L., 2018. Photochemical model evaluation of 2013 California wild fire air quality impacts using surface, aircraft, and satellite data. Science of The Total Environment 637, 1137-1149.

Zhou, L., Baker, K.R., Napelenok, S.L., Pouliot, G., Elleman, R., O'Neill, S.M., Urbanski, S.P., Wong, D.C., 2018. Modeling crop residue burning experiments to evaluate smoke emissions and plume transport. Science of The Total Environment 627, 523-533.

---

## Referee Comment (RC2) · Anonymous Referee #2 · 12 May 2019

The paper presents a number of updates to the Canadian operational biomass burning (BB) emissions model and its verification for North America (NA) for 2017 fire season. Several improvements have been made to the model to improve the parameterizations of the BB emissions, fire plume rise and behavior. These updates have resulted in improvement of the simulated O3 and PM2.5 concentrations.

The development of new capabilities for smoke forecasting is very important. As recent years showed the wildfires in the US and Canada can cause severe air pollution episodes affecting millions of people. Accurate and timely air quality forecasting plays a

critical role for stakeholders and public to mitigate the effects of the adverse air pollution from wildfires.

The paper is well organized. This study deserves to be published in GMD. My major comment is that while the verification of the ground level PM2.5, O3 and NO2 are important, it is uncertain how accurately the model simulates concentrations of the chemical species aloft. Smoke aerosols in the atmosphere affect radiation, thus affecting weather and climate. The authors demonstrate that the new plume rise algorithm injects fire emissions at higher altitudes compared to the previous version of the model. This change leads to the reduction of the high bias in the ground level PM2.5 concentrations forecast by the older model. To verify the PM2.5 concentration simulations within entire atmospheric column, it would be helpful to compare the model predicted AOD fields. Figure 10 illustrates the model's ability in capturing the wide smoke plume over NA. However, this is a qualitative comparison. I realize that a full quantitative verification of the model versus the satellite AOD is beyond scope of the paper. Therefore, I suggest comparing the model with satellite measured AOD over NA for 1-2 episodes at least, so a reader can get an idea how realistic is the forecast total aerosol burden from fires.

Page 18. If you discuss these SI figures here, then move to the main text.

Table 7. Are these daily concentrations? Specify.

I suggest merging section 4 and 5.

---

## Author Comment (AC1) · 17 Jun 2019

**General responses from the authors**

The authors thank both referees for their time and effort in reviewing this manuscript. Their suggestions were very helpful in improving the paper.

We address individual referee's comments below, in blue font, following the original comments indicated by italics.

Please note that all table and figure numbers referenced in our responses are based on those in the original manuscript and original Supplementary Material (SM). However, we have made a number of revisions to both documents (attached). Aside from textual changes, we also moved Figure S5 from the SM to the end of Section 3.1.5 of the main text as Figure 11, and we have added four new tables, Tables S2, S4, S5, and S10, to the SM.

**Anonymous Referee #1:**

➢ *The authors compare modeling system predictions of PM2.5, O3, and NO2 against routine surface measurement sites from 3 different forecasting systems: 1) a system with no wildfire, 2) the existing version of the Canadian forecasting system, and 3) the newly updated Canadian forecasting system. Simulations 2 and 3 include wildfire emissions, which are treated differently in each system. Multiple enhancements to simulating wildfire emissions were implemented in the new forecasting system compared to the existing system including emission factors, plume height, and vertical distribution of smoke emissions within the plume.*

*Wildfire impacts are important to provide so that the public can be better informed about potential or actual exposure to change behavior to try and minimize exposure. The work presented here is very relevant and important for the broader modeling community and providing confidence in the approach is important toward establishing confidence in the forecasting product being provided to the public to understand reliability in the forecast. Overall, the changes made to the approach for estimating wildfire emissions make sense and would seem to favor an improved forecast. It is not totally clear which changes make the most impact since they were not systematically evaluated. However, the main concern is that it is not clear based on the information provided how well the system is performing for capturing smoke impacts. This is important work and very useful information that should be published. Comments that follow are intended to provide a stronger evaluation to more clearly isolate how the improvements in the forecasting system improve performance, which is a challenge since many monitors will typically be largely impacted by non-wildfire sources.*

Thank you for your positive comments. See below for our responses to your specific comments.

➢ *Too many monitor locations and too long of a period were aggregated, which really limits our understanding of how much better the new system is at predicting smoke impacts on PM2.5, O3, and NO2. A clearer demonstration (maybe in addition to the broad evaluation provided) would be to focus on a specific area of monitors that were known to be impacted for a few days or weeks by wildfire and see how well each of the modeling systems capture this known episode.*

We struggled with this exact issue. It is difficult to balance the scale of model evaluation in this study as the FireWork system is an operational product with a model domain covering North America. We wanted to demonstrate that the science changes implemented in this work provide an overall improvement to model forecast skill, while at the same time not degrading forecast results in areas not impacted by wildfires. This overall model forecast evaluation throughout the fire season at the continental scale was presented in Section 3.1.3 for the 2017 fire season and in the SM for the 2018 season (Tables S7 and S8).

We also focused on smaller areas and shorter time periods with greater and more frequent wildfire impacts. We provided a regional-scale analysis in Section 3.1.4 on model forecast performance for three separate regions (AB+BC, ID+MT+OR+WA and Northern-Canada) in August 2017, and in the SI (Figures S9-S11) for August 2018 for two provinces (BC, AB) and two states (WA and MT).

Lastly, although it was not discussed in the main paper, we provided several episode analyses for localized, station-based PM$_{2.5}$ surface-level, hourly time series comparisons as part of a subjective evaluation across many stations (in Section 3.1.5). These analyses considered two 4-day intensive periods in Aug. 12-15, 2017 (Figure S5) and Aug. 22-25, 2018 (Figure S13) when pronounced fire impacts were evident. To increase the visibility of these station-based episodic comparisons, we have moved Figure S5 from the SM to the main text (now Figure 11).

> *Another approach would be to provide the monitor comparisons only when the model is predicting smoke impacts (whether either of the systems with wildfire emissions predicted smoke impacts). The way the information is presented it is not clear at all how well each system captures fire impacts, in particular for O3 and NO2. This is because the model performance is likely dominated by other sources or regions and not by the wildfire and the new approaches for estimating smoke may simply be compensating for underrepresentation of other sources. So if the authors can focus on monitors with periods dominated by wildfire impacts that would help make the evaluation much clearer.*

The approach suggested is conditional on being able to identify stations and analysis periods that are positively impacted by wildfire smoke. One drawback of the approach is thus that it can miss events that were not forecasted by the model, and hence it may artificially enhance the model performance metrics as only stations modelled to be impacted by fire plumes are utilized in the calculation (i.e., only correct events and false positives are considered).

Nevertheless, this is a good suggestion and we have conducted one such evaluation for the period Aug. 1 to Sept. 18, 2017 when fire activity was high in western Canada and the western U.S. We had already presented results for this period in Tables 8, S3, and S5, but this time only stations and days with observed daily maximum $PM_{2.5}$ concentrations greater than 50 µg m$^{-3}$ were considered. This threshold was selected to be more than double the mean modelled $PM_{2.5}$ concentrations without fire emissions (RAQDPS) and it is used as a proxy for selecting stations and times that were influenced by forest fire plumes.

The following table shows model performance statistics are in general agreement with analysis that considered all measurement stations within the region, but with a stronger signal due to preselection of days and stations that had high observed $PM_{2.5}$ concentrations.

Model performance statistics for daily maximum $PM_{2.5}$ (µg m$^{-3}$), $O_3$ (ppbv) and $NO_2$ (ppbv) for stations and days where observed daily maximum $PM_{2.5}$ is greater than 50 µg m$^{-3}$ during high fire activity period of Aug. 1 to Sept. 18 2017. Only stations within the 2 regions of interest are analyzed in the grouping:

| Species | $PM_{2.5}$ | | | $O_3$ | | | $NO_2$ | | |
|---|---|---|---|---|---|---|---|---|---|
| Stations | 160 (AB+BC:72, WA+OR+ID+MT:88) | | | 67 (AB+BC:58, WA+OR+ID+MT:9) | | | 62 (AB+BC:61, WA+OR+ID+MT:1) | | |
| Model | RAQDPS | FWops | FWcffeps | RAQDPS | FWops | FWcffeps | RAQDPS | FWops | FWcffeps |
| Count | 1772 | | | 590 | | | 519 | | |
| Ō | 95 | | | 52 | | | 19 | | |
| M̄ | 20 | 289 | 146 | 63 | 151 | 74 | 23 | 30 | 23 |
| MB | -76 | 194 | 50 | 12 | 100 | 22 | 4 | 11 | 4 |
| R | -0.04 | 0.36 | 0.46 | 0.45 | 0.04 | 0.42 | 0.59 | 0.51 | 0.58 |
| RMSE | 91 | 533 | 179 | 27 | 156 | 36 | 18 | 30 | 19 |

We have now included this table in the SM as Table S10 and we have added the following text referring to it to the end of Section 3.1.4:

"One additional evaluation was conducted for the Aug. 1 to Sept. 18, 2017 period to examine model performance for just those stations and days observed to be affected by wildfire plumes. Table S10 presents performance statistics for model predictions of daily maximum $PM_{2.5}$, $O_3$, and $NO_2$ concentrations for a filtered subset of measurement for which observed daily maximum $PM_{2.5}$ levels at individual stations were above 50 µg m$^{-3}$. Table S10 can be compared with Tables 8, S6, and S8, but it includes only 22% of the daily maximum $PM_{2.5}$ measurements, 14% of the daily maximum $O_3$ measurements, and 15% of the daily maximum $NO_2$ measurements considered in those three tables. Although both observed and modelled values are higher in Table S10 than the other three tables, the ranking of relative model performance is in general agreement with the analyses that considered all measurement stations within the regions and all days in the evaluation period."

➢ *The literature review in the introduction misses a few newer papers that looked at the performance of the Briggs approach. In these papers the Briggs approach performed reasonably for wildfire when provided realistic information about the fire size and timing. It does not appear that the approach limits plume tops to the modeled PBL height or uniformly distributes emissions through the plume. These may simply be choices made by the model developers for their implementation of Briggs but it begs the question about why these limitations were imposed in this system (or the older version of the system). Briggs is a component of the CMAQ model used for air quality forecasts by NOAA so references for that system may be relevant for the modified Briggs approach (Baker et al., 2018; Zhou et al., 2018) and a very brief note that the modified Briggs has been implemented differently in other systems.*

This is a valid point and we have made changes to the manuscript to indicate that the implementation of Briggs plume rise parameterization can be different in different modeling systems.  We have also updated the literature review to include the work using the CMAQ model that showed adequate results with Briggs parameterization in the treatment of forest fire plumes.  Specifically:

> Section 1, paragraph 7:
> "… Furthermore, different interpretations and implementations of plume-rise parameterizations within CTMs can also result in differences in modelled plume injection heights for both facility stacks and fire sources.  For example, recent model experiments using the CMAQ model showed that a modified Briggs parameterization with estimated fire buoyancy heat flux can adequately capture plume injection heights from wildfires and prescribed fires (Baker et al., 2018; Zhou et al., 2018)."

> Section 3.1.2, paragraph 3:
> "… In a recent study on model plume-rise parameterization, Akingunola et al. (2018) demonstrated that the current implementation of the Briggs scheme in GEM-MACH under-predicts measurements from facility stacks and can be further improved with a layered lapse-rate approach that is not currently used in the RAQDPS…"

➢ *Using 2010 Canadian emissions and 2011 U.S. emissions to represent 2017 (and later) will result in an overestimation of pollutants due to significant reductions in mobile source emissions… during this period, plant shutdowns, and fuel switching by EGUs from coal to natural gas. The emissions should be updated for the forecasting system and adjusted to reflect broader sector reductions.  For the purposes of this manuscript and evaluation, the authors do not need to re-do the entire evaluation with newer emissions but need to clearly recognize that the emissions are likely leading to large overestimates of some species (like NOX in particular) which confounds the evaluation since wildfire NOX may not make performance "look better".*

Both FireWork-Ops and FireWork-CFFEPS include the same full suite of anthropogenic emissions as used by the RAQDPS system without wildfire emissions.  The RAQDPS is ECCC's operational regional AQ forecast system throughout the year.  The RAQDPS uses official emissions inventory data from Canada, the U.S. and Mexico.  The special emissions-processing-ready versions of these inventories needed to prepare gridded emission files for use by AQ models are typically available only every 3 to 5 years.  As noted by the referee, the forecast model versions considered in this study used older official emissions inventories for Canada (2010) and the U.S. (2011), but we failed to note recent updates to these operational emissions implemented in 2018 (Moran et al., 2018).

To understand the magnitudes of the anthropogenic emission changes between 2010 and 2017 in western North America, where most wildfires occur, we have compared inventory emissions for the base years used by the RAQDPS and FireWork with 2017 emissions for the regions of interest (ID+MT+OR+WA and AB+BC).  The following table shows the actual and relative changes in emissions by region and species.

| (tonnes) | CO | | NOx | | PM$_{2.5}$ | | SO$_2$ | | VOC | |
|---|---|---|---|---|---|---|---|---|---|---|
| | 2011 | 2017 | 2011 | 2017 | 2011 | 2017 | 2011 | 2017 | 2011 | 2017 |
| ID+MT+OR+WA Total | 4,165,074 | 3,413,646 | 574,066 | 393,988 | 380,461 | 383,660 | 84,307 | 49,872 | 899,032 | 815,470 |
| Relative change | | -18% | | -31% | | +1% | | -41% | | -9% |

| (tonnes) | CO | | NOx | | PM$_{2.5}$ | | SO$_2$ | | VOC | |
|---|---|---|---|---|---|---|---|---|---|---|
| | 2010 | 2017 | 2010 | 2017 | 2010 | 2017 | 2010 | 2017 | 2010 | 2017 |
| BC+AB Total | 1,976,489 | 1,749,174 | 1,013,989 | 932,227 | 605,857 | 674,816 | 462,687 | 318,534 | 784,144 | 724,425 |
| Relative change | | -12% | | -8% | | +11% | | -31% | | -8% |

It can be seen that NOx emissions decreased by 8% over this period in western Canada and by 31% in the Pacific Northwest, whereas primary PM$_{2.5}$ emissions were little changed. As can be seen in Figures 8 and 9, in regions affected by wildfires, PM$_{2.5}$ levels are completely dominated by wildfire emissions and anthropogenic emissions trends have very little impact. However, the referee has raised an important issue and we have now added this table to the SM as a new Table S2 to quantify the impact of using older emission inventories for AQ forecasting.

The paragraph in the manuscript related to the anthropogenic emissions considered in this study (Section 2.1, paragraph 3) has been revised accordingly. It now reads:

> "Emission files used by the RAQDPS include emissions from both anthropogenic and biogenic sources. The anthropogenic emission inventories that are considered are updated every few years. For the 2017 operational runs considered here, RAQDPS anthropogenic emission files were based on the 2010 Canadian national Air Pollutant Emissions Inventory (APEI), the 2011 U.S. National Emissions Inventory (NEI), and the 1999 Mexican NEI. These inventories were processed using the SMOKE emissions processing system to generate files of hourly, gridded, chemically-speciated emissions fields (Zhang et al., 2018). Biogenic emissions are calculated online in the RAQDPS based on the algorithm from BEIS version 3.09 with BELD3-format vegetation land cover for Canada and the U.S. It is worth noting that the RAQDPS anthropogenic input emissions were updated in Sept. 2018 based on the 2013 Canadian APEI, a projected 2017 U.S. NEI, and the 2008 Mexican NEI (Moran et al., 2018). In order to understand the impact of using inventories for older base years, Table S2 compares 2010/2011 and 2017 inventory values for several western Canadian provinces and northwestern U.S. states. Over this period, NO$_x$ and VOC emissions decreased by 8% and 8% in western Canada and by 31% and 9% in the northwestern U.S., respectively, whereas PM$_{2.5}$ emissions increased by 11% and 1%. The actual magnitudes of these differences are comparable to or smaller than the estimates of NO$_x$, VOC, and PM$_{2.5}$ emissions from North American wildfires given in Table 5."

➢ *It is not clear which emissions are used for each system. Are fuel and combustion specific emissions used in any of the simulations? Text suggests they may be in the analysis but other text and the Tables suggest otherwise. What is the source of the emission factors?*

The FireWork-Ops system, which is described in Section 2.1, calculates fire emissions using emission factors from the FEPS system. The new FireWork-CFFEPS system, described in Section 2.2, calculates fire emissions more dynamically with updated emission factors. The new emission factors used in FireWork-CFFEPS (Table 3) are based on the paper by Urbanski (2014). In both systems, the fuel burned and the relative length of the combustion stages are dependent on fuel type (Tables 1 and 2) but emission factors are not dependent on fuel type (Table 3). This latter option is available in the FireWork-CFFEPS system, but it was not implemented in the current study. This assumption is noted in Section 2.2.2 (last paragraph), Section 2.3.4 (first paragraph; note small changes), and Section 4:

> "…While FireWork-Ops uses average emission factors from FEPS, updated emission factors were chosen for CFFEPS based on recent literature (Urbanski, 2014)… These emission factors are applied for all input fuel types in the current application, although CFFEPS is now designed to allow for fuel-specific values as found in recent measurements (Liu et al., 2017)…"

"Although emissions factors can also be dependent on fuel type, current input has one default emission factor applied to all fuel types (Table 3)"

"The same emission factors are now applied for all input fuel types in CFFEPS, but emission factors can vary by fuel type as found in recent measurements (Liu et al., 2017) and fuel-type-specific emission factors can be considered in future."

➢ *Please provide the speciation profiles for the different combustion components for VOC and PM2.5 in the supporting information with a citation as that is important and would be valuable to others modeling wildfire.*

The chemical speciation profiles for NMHC and $PM_{2.5}$ emissions from wildfires used in FireWork-CFFEPS are based on speciation profiles from the EPA's SPECIATEv4.5 database. This is described in Section 2.3.4. These speciation profiles and their ADOM-2 mechanism species descriptions are now provided in the SM as new Tables S4 and S5.

---

## Author Comment (AC2) · 17 Jun 2019

**General responses from the authors**

The authors thank both referees for their time and effort in reviewing this manuscript. Their suggestions were very helpful in improving the paper.

We address individual referee's comments below, in blue font, following the original comments indicated by italics.

Please note that all table and figure numbers referenced in our responses are based on those in the *original* manuscript and Supplementary Material (SM). However, we have made a number of revisions to both documents (attached). Aside from textual changes, we also moved Figure S5 from the SM to the end of Section 3.1.5 of the main text as Figure 11, and we have added four new tables, Tables S2, S4, S5, and S10, to the SM.

**Anonymous Referee #2:**

➢ *The paper presents a number of updates to the Canadian operational biomass burning (BB) emissions model and its verification for North America (NA) for 2017 fire season. Several improvements have been made to the model to improve the parameterizations of the BB emissions, fire plume rise and behavior. These updates have resulted in improvement of the simulated O3 and PM2.5 concentrations.*

*The development of new capabilities for smoke forecasting is very important. As recent years showed the wildfires in the US and Canada can cause severe air pollution episodes affecting millions of people. Accurate and timely air quality forecasting plays a critical role for stakeholders and public to mitigate the effects of the adverse air pollution from wildfires.*

*The paper is well organized. This study deserves to be published in GMD.*

Thank you for your positive comments.

➢ *My major comment is that while the verification of the ground level PM2.5, O3 and NO2 are important, it is uncertain how accurately the model simulates concentrations of the chemical species aloft. Smoke aerosols in the atmosphere affect radiation, thus affecting weather and climate. The authors demonstrate that the new plume rise algorithm injects fire emissions at higher altitudes compared to the previous version of the model. This change leads to the reduction of the high bias in the ground level PM2.5 concentrations forecast by the older model. To verify the PM2.5 concentration simulations within entire atmospheric column, it would be helpful to compare the model predicted AOD fields. Figure 10 illustrates the model's ability in capturing the wide smoke plume over NA. However, this is a qualitative comparison. I realize that a full quantitative verification of the model versus the satellite AOD is beyond scope of the paper. Therefore, I suggest comparing the model with satellite measured AOD over NA for 1-2 episodes at least, so a reader can get an idea how realistic is the forecast total aerosol burden from fires.*

We agree that a more systematic assessment of model evaluation is desirable, especially for species concentrations aloft where changes to the plume-injection-height parameterization will have the largest impacts. However, the main goal of the operational FireWork system is to provide numerical guidance on air quality conditions to regional forecasters and emergency first-responders, for whom pollution episode arrival time and surface-level concentrations are the most important forecast quantities. Surface-level model performance metrics are thus the focus of this work.

Additional research is currently underway to evaluate the model's upper-air performance and plume-injection height parameterization using a research version of the GEM-MACH model with a 12-bin aerosol size representation. Model results are being analyzed against satellite plume-height retrievals from the MISR sensor and aircraft measurements of a wildfire from a measurement campaign that took place in Alberta in July 2018. This work is alluded to in Section 4 (2nd, 5th, and 6th paragraphs). In addition, AOD estimates made from the 12-bin GEM-MACH model are better suited for these analyses than AOD estimates from the 2-bin GEM-MACH model used in the operational forecast evaluation experiments presented here, where operational bin 1 covers the diameter size range from 0 to 2.5 μm.

Given that the focus of the paper is on the first-step "operational evaluation" of the new AQ forecast system where surface level results are critical, and that the paper is already lengthy, we would prefer to save a more detailed, upper-air analysis for a next, follow-up study. As you noted, though, we have provided a subjective VCD comparison with satellite imagery as a proxy subjective evaluation for total column $PM_{2.5}$ in Sec. 3.1.5, as this imagery is part of the forecast product suite from the operational FireWork system. In addition, we have moved Figure S5, which shows a comparison of both satellite imagery and surface measurements at individual stations with predictions from the two FireWork versions, from the SM to the end of this section (new Figure 11).

➢ *Page 18. If you discuss these SI figures here, then move to the main text.*

Thank you for the suggestion. We have chosen not to implement this change because we were concerned that the additional tables and figures for the 2017 $O_3$ and $NO_2$ evaluations (Tables S3-S6 and Figures S1-S4) might distract from the $PM_{2.5}$ model forecast results, which we think are more important given wildfire $PM_{2.5}$ impacts on human health and visibility. Furthermore, our analysis showed that most significant changes are in $PM_{2.5}$ predictions as compared to $O_3$ and $NO_2$ predictions.

➢ *Table 7. Are these daily concentrations? Specify.*

Thank you for pointing this out. Table 7 shows categorical scores (POD, FAR, CIS) for $PM_{2.5}$/$O_3$/$NO_2$ over regions of interest for the 3 months in 2017. These scores are calculated based on hourly modelled and measurement values paired by grid location and time. These categorical scores were defined in Section 3, and we have modified the description in the paper at the beginning of Section 3 to specify clearly that hourly values were used in these calculations:

> "… Model hourly results for the near-surface concentrations at measurement site locations were extracted, paired by time, and evaluated using common model evaluation statistics as well as three operational, forecast-oriented categorical scores (Jolliffe and Stephenson, 2012). The categorical scores, calculated from hourly values, were probability of detection (POD), false alarm ratio (FAR), and critical success index (CSI), where:…"

We have also modified the Table 7 caption to indicate that hourly values were used.

➢ *I suggest merging section 4 and 5.*

Sections 4 and 5 cover different topics and are naturally separate. In the current layout, Section 4 highlights outstanding development issues not currently considered in FireWork-CFFEPS system, and emphasis future evaluations that may better attribute changes of forecast results to model developments. Section 5 provides an overall summary and conclusions of the work and can be useful for readers who want a quick synopsis of the study.

As the paper is already lengthy, with many figures and tables, we believe that keeping these sections separate improves the overall readability of the paper.